# Resonant neutron reflectometry for hydrogen detection

L. Guasco [1,2], Yu. N. Khaydukov [1,2], S. Pütter [3], L. Silvi [4], M. A. Paulin [4,5], T. Keller[1,2] & B. Keimer [1✉]

The detection and quantification of hydrogen is becoming increasingly important in research on electronic materials and devices, following the identification of the hydrogen content as a potent control parameter for the electronic properties. However, establishing quantitative correlations between the hydrogen content and the physical properties of solids remains a formidable challenge. Here we report neutron reflectometry experiments on 50 nm thick niobium films during hydrogen loading, and show that the momentum-space position of a prominent waveguide resonance allows tracking of the absolute hydrogen content with an accuracy of about one atomic percent on a timescale of less than a minute. Resonance-enhanced neutron reflectometry thus allows fast, direct, and non-destructive measurements of the hydrogen concentration in thin-film structures, with sensitivity high enough for real-time in-situ studies.

[1] Max-Planck-Institut für Festkörperforschung, Heisenbergstraße 1, D-70569 Stuttgart, Germany. [2] Max Planck Society Outstation at the Heinz Maier-Leibnitz Zentrum (MLZ), D-85748 Garching, Germany. [3] Forschungszentrum Jülich GmbH, Jülich Centre for Neutron Science (JCNS) at Heinz Maier-Leibnitz Zentrum (MLZ), Lichtenbergstr. 1, D-85747 Garching, Germany. [4] Helmholtz Zentrum Berlin, Hahn-Meitner-Platz 1, 14109 Berlin, Germany. [5] Laboratorio Argentino de Haces de Neutrones, CAB, CNEA, R8402AGP Bariloche, Argentina. ✉email: b.keimer@fkf.mpg.de

In most of the technologies underlying the vision of a "hydrogen economy", hydrogen is present either within or in immediate proximity to materials. Sensitive methods to quantify the incorporation of hydrogen in solids are hence of intense interest across a wide variety of research fields ranging from fuel cells[1] to photocatalysts[2] and fusion reactors[3]. A rapidly developing research frontier is taking advantage of hydrogen intercalation to modify the electronic properties of solids and solid-state devices. Prominent examples include targeted modification of the lattice architecture and doping level of quantum materials,[4–8] modulation of the exchange coupling and magnetic anisotropy of magnetic multilayers and devices,[9–15] and solid-state gas sensors[16,17]. As even small amounts of hydrogen can elicit a substantial electronic response, the role of unwanted residuals of hydrogenous reagents for topotactic materials modification has also been prominently discussed[18,19].

Understanding and controlling these effects require quantitative information about the hydrogen concentration inside materials. Thanks to the large neutron scattering power of hydrogen, neutron imaging has become a powerful, direct characterization method for hydrogen in bulk materials[20]. Thin films are particularly suited for quantitative investigations of the influence of hydrogen on different physical properties of materials, because homogeneous profiles of both hydrogen and the quantity of interest can be easily obtained, and measurements can be carried out in well-defined geometries. For thin films, neutron reflectometry (NR) allows quantification and depth profiling of H atoms[21,22]. By analogy with the optics of light and X-rays, NR measures the ratio of reflected and incoming intensities (the reflectivity $R$) as a function of momentum transfer, $Q$. Experimental $R(Q)$ curves are then modeled to extract the depth dependence of the "neutron optical potential" $\rho(z)$. Injection of hydrogen in the studied sample leads to a modification of $\rho(z)$, which can be traced via the altered reflectivity. Hydrogen concentrations of 5 at.% can be reliably measured by conventional NR[23–26]. Smaller quantities of H are still detectable[23] and can be studied in real-time if the kinetics of absorption is slow enough[21,23,24], and as we will show below, there is some scope for sensitivity enhancement of NR via optimization of the experimental conditions. Yet, as real-time experiments remain severely limited by the required exposition times and beamtime at high-intensity neutron sources, additional methodological advances are required to fully realize the potential of neutron reflectometry in this rapidly developing field of research.

We have taken advantage of waveguide resonances resulting from the formation of neutron standing waves in thin films[27] to substantially increase the sensitivity of neutron reflectometry for real-time hydrogenation experiments. We demonstrate this technique by reporting in situ experiments on Nb films during hydrogen loading, which show that fast hydrogenation (~0.5 at.% per 60 s) at room temperature can be followed with a sensitivity of ~1 at.%. The neutron data correlate very well with the electrical resistivity measured simultaneously. Resonance enhancement thus expands the application range of specular NR, while retaining some of its unique advantages. In particular, neutron methods yield absolute measurements of the hydrogen concentration (in contrast to X-ray reflectometry or diffraction, which monitor the hydrogenation process indirectly via its impact on the layer thickness[28–30]), and radiation damage is negligible (in contrast to nuclear methods that require MeV-energy ions[31,32]). Another advantage of neutron-based techniques is the possibility to perform in situ studies, while nuclear methods often require high vacuum. With its high sensitivity and its ability to study fast kinetics, resonant neutron reflectometry (RNR)—as well as other methodological advances—therefore have the potential to develop into powerful tools for the microscopic understanding and control of hydrogen effects in solids.

## Results

**Neutron waveguide resonances.** Before presenting measurements that demonstrate waveguide resonances in thin-film structures and their sensitivity to hydrogenation, we briefly review the mechanisms underlying their formation[27]. We consider a generic three-layer potential comprising a capping layer, an active layer, and a substrate (layer index $i = 1–3$, respectively). Each layer is characterized by the corresponding scattering length density (SLD) $\rho_i$ and thickness $d_i$. The amplitude of the neutron wavefunction $\Psi$ in the central layer can be calculated by solving the neutron's one-dimensional Schrödinger equation:

$$\Psi = \frac{t \times e^{ik_1 d_1}}{1 - r_{21} r_{23} e^{2ik_2 d_2}}. \tag{1}$$

Here $t$ is the transmission amplitude of neutrons through the capping layer, and $r_{ij}$ are the Fresnel amplitudes of a single reflection at the interface between two consecutive layers:

$$r_{ij} = \frac{k_i - k_j}{k_i + k_j}, \tag{2}$$

where $k_0$ and $k_i = (k_0^2 - 4\pi\rho_i)^{1/2}$ are the normal components of the neutron wavevector in vacuum and in the $i$th layer, respectively. The most distinctive property of the Fresnel amplitudes is the total reflection ($|r_{ij}| = 1$) that takes place at the $i/j$ interface when $k_0 < k_0^{\text{crit}} = \sqrt{4\pi(\rho_j - \rho_i)}$. In this regime the amplitude of the wavefunction shown in Eq. (1) can be enhanced under the following conditions:

$$|r_{23}| = |r_{21}| = 1 \tag{3a}$$

$$\arg(r_{21}) + \arg(r_{23}) + 2\,\text{Re}\,(k_2)d_2 = 2\pi n \tag{3b}$$

with $n$ integer. For condition (3a) to hold, a strong gradient of $\rho$ must be present at the interfaces around the central layer. Various SLD profiles can provide the requisite contrast, including potential wells, steps, and staircases.

In general, determining the wavevector transfer of the waveguide resonance, $Q_{\text{res}} = 2k_{\text{res}}$, requires a numerical solution of Eq. (3b). Here we are interested in the shift of $Q_{\text{res}}$ by hydrogen incorporation into the active layer, which can be obtained analytically (following ref. [33]) if the resulting change in SLD, $\Delta\rho_2$, is small, and changes in the layer thickness $d_2$ can be neglected:

$$\Delta Q_{\text{res}} \cong \frac{8\pi\Delta\rho_2}{Q_{\text{res}}^0} = \frac{8\pi\bar{N}}{Q_{\text{res}}^0} b_H c_H \tag{4}$$

where $Q_{\text{res}}^0$ is the resonance position in the pristine structure, $\bar{N}$ is the average atomic density of the active layer, and $b_H$ and $c_H$ are the scattering length and atomic concentration of hydrogen in the active layer, respectively. The resonance position thus depends linearly on the hydrogen concentration, with a proportionality constant comprising parameters that are either known or can be measured straightforwardly. Measuring the shift of the waveguide resonance upon hydrogenation thus yields the absolute hydrogen concentration for small $c_H$. Accurate measurements for large $c_H$ require additional knowledge of the H-induced expansion of $d_2$ (see below).

Resonance results from trapping neutrons inside the waveguide layer and does not manifest itself in a straightforward way. For its detection, a channel is required to withdraw neutrons from the main channel with a scattering cross section that depends on the neutron density $|\Psi|^2$. Possible options include neutron absorption with detection of secondary radiation such as $\gamma$[34] or $\alpha$[35] particles from reactions (n, $\gamma$) and (n, $\alpha$), incoherent scattering from the H atoms[36], neutron channeling[37], or transmission[38]. The position of the resonance will correspond to a peak in the secondary channel

and a dip on the reflection plateau. For our study we have chosen spin-flip scattering from a magnetic label layer with magnetization aligned non-collinearly to the direction of neutron polarization, because this method is available at many neutron reflectometers all over the world, allows measurements with particularly high signal-to-background ratio, is theoretically well understood[39], and has been experimentally established in different systems[36,40,41]. In addition, the chosen design will allow one to investigate in the future the effect of hydrogen on the magnetic state of thin films, a promising direction in spintronics[15]. A disadvantage of using a magnetically noncollinear label is the necessity to polarize the neutron beam, which leads to the loss of at least a factor of two in the incoming intensity. However, as will be shown in the following sections, it still allows for a faster measurement of hydrogen loading with respect to classical NR.

**Sample design and measurement scheme.** For our proof-of-principle experiment, we used molecular beam epitaxy to grow films with a 50 nm thick active layer of niobium, a well-known hydrogen absorber[42], on sapphire substrates (see Methods for details of the synthesis procedure and Supplementary Materials for detailed characterization of the samples). The diffusion coefficient of hydrogen in niobium is only one order of magnitude lower than typical self-diffusion coefficients of liquids[43], thus enabling fast loading and homogeneous distribution of hydrogen throughout the entire active layer. The sapphire substrate was chosen because it exhibits a significant SLD contrast with the active overlayer ($\rho = 5.72$ versus $3.91 \times 10^{-6} \text{Å}^{-2}$). As magnetic label layer we used a 3 nm thick Co layer inserted in the middle of the Nb layer (Fig. 1a). The choice of material, thickness, and position of the label layer was dictated by the following considerations. The layer is thin enough not to distort the waveguide properties of the system (Fig. 1c), but thick enough to be ferromagnetic with a moment approaching the bulk value (Supplementary Fig. 1d). We also expect that the thin Co layer has a high transmittance for hydrogen atoms, in analogy to the Fe layers used in prior hydrogenation experiments on thin films and multilayers[10,11,36]. In contrast to Fe, however, Co is known to form sharp interfaces with Nb[44,45]. Finally, we placed the label layer in the middle of the structure for the sake of proof-of-principle demonstration. Alternatively, one may use a label layer on top of the active layer or on the top of the substrate, as demonstrated in ref. [36]. The structure was capped by a 3 nm thick Pt layer to initiate splitting of $H_2$ molecules at the surface and diffusion of H atoms into the active layer[46].

Figure 1b shows the calculated scattering potential of the sample in its pristine state and after a loading with 85 at.% hydrogen, which reduces the SLD of the active layer to $\rho = 2.13 \times 10^{-6} \text{Å}^{-2}$. We note that for illustration purposes, we have neglected the H-induced swelling of the Nb layer, which amounts to ~10% at this level of concentration[21,30]. As Co is much less affine to hydrogenation, the SLD of the Co layer is expected to remain unaltered (as shown in the case of Fe/Nb[21]). Figure 1c shows the local neutron density $|\Psi|^2$ as a function of depth and $Q$ calculated for these profiles. In both cases, the neutron density enhancement is maximal at the depth of the Co label layer, ensuring maximal sensitivity to the resonant field. In its virgin state, the waveguide resonance amplifies the neutron density by a factor of 16. Loading with 85 at.% hydrogen induces a ~20% shift of the resonance, and a ~40% reduction of its intensity.

**Polarized neutron reflectometry.** Figure 2a shows the polarization-resolved neutron reflectivity of the as-prepared

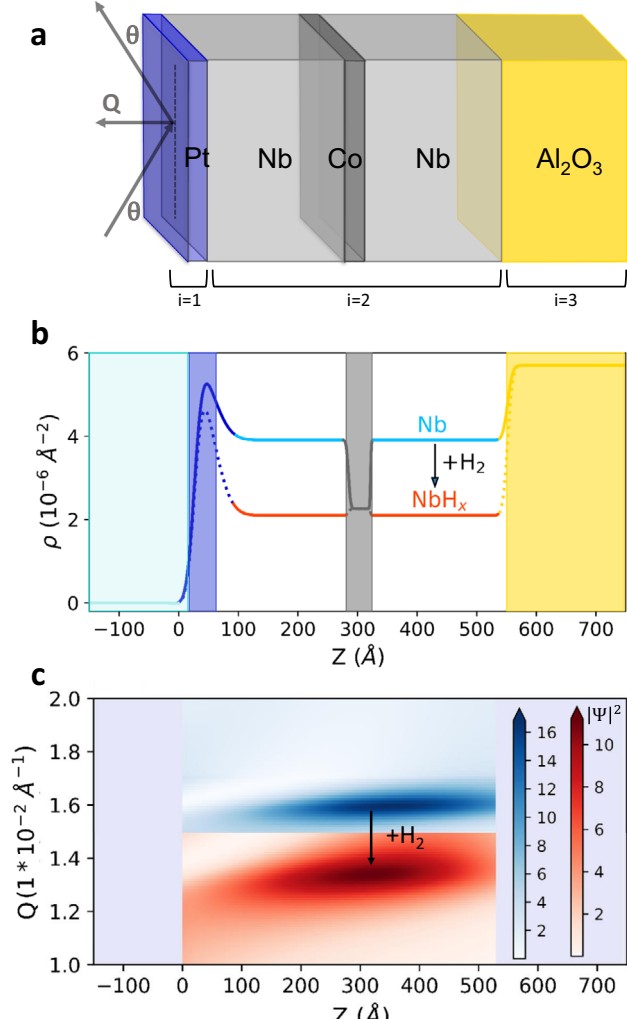

**Fig. 1 Resonant neutron reflectometry scheme. a** Sketch of the thin-film structure and geometry of the reflectometric measurements. $Q$ is the wavevector transfer, and $\theta$ the scattering angle. **b** Depth profile of the scattering length density $\rho$ of the sample before (blue line) and after (red line) incorporation of 85 at.-% hydrogen. **c** The numerically calculated neutron density enhancement (with respect to the incoming beam with $|\Psi|^2 = 1$), along the film before (blue) and after (red) hydrogenation. For simplicity, the schematic does not include the H-induced swelling of the film.

sample in the non-spin-flip ($R^{++}$ and $R^{--}$) and spin-flip ($R^{-+}$ and $R^{+-}$) channels. The data shown are well described by reflectivity simulations based on the exact solution of the Schrödinger equation for a neutron moving in the potential imposed by the nuclear and magnetic SLD profiles shown in the inset of Fig. 2b. The parameters characterizing these profiles were adjusted to yield the best fit to the experimental data. The thicknesses and SLDs resulting from these fits are in good agreement with the nominal values, and the interfacial roughnesses are well below the layer thickness (see Methods and Supplementary Materials). A prominent waveguide resonance is observed in the spin-flip channels (dashed line in Fig. 2). Its position, width, and intensity are well described by the simulations (solid lines in Fig. 2).

We now proceed to in situ measurements of the waveguide resonance during exposure to $H_2$ gas at room temperature. The RNR method was applied to two simultaneously grown samples (samples 1 and 2 in Fig. 3) in two different modes. For sample 1,

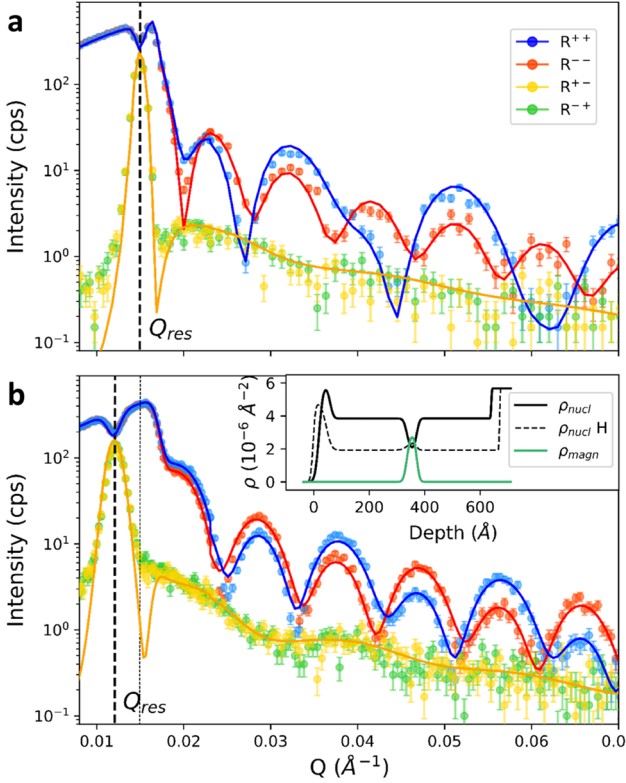

**Fig. 2 Polarized neutron reflectivity before and after hydrogen absorption. a** as-prepared sample and **b** fully hydrogenated sample. The data were taken at the NREX reflectometer at room temperature in a magnetic field of 5 Oe perpendicular to the scattering plane. The dashed lines mark the position of the waveguide resonance in the spin-flip channels, $Q_{res}$. The solid lines are the results of fits described in the text. The inset shows the corresponding nuclear and magnetic scattering length profiles. Error bars indicate the statistical error.

we tracked the shift of the resonance by measuring a series of consecutive $Q$-scans around the resonance position during hydrogen loading, as shown in Fig. 3a. The time dependence of the peak maximum (Fig. 3b, left scale) can be straightforwardly translated into the corresponding absolute hydrogen concentration, $c_H$, using Eq. (4) (Fig. 3b, right scale).

For the second sample (sample 2 in Fig. 3), the shift of the resonance peak was tracked by monitoring the count rate at fixed-$Q$ during H-loading, as shown in Fig. 3e. To maximize the sensitivity of RNR, we chose the $Q$-value corresponding to the maximum derivative of the resonant reflectivity in vacuum ($Q^*$ in Fig. 3d). To calibrate the new method against a standard, indirect H-detection method, we simultaneously measured the resistivity of the sample, which is known to depend on the hydrogen content[47]. Figure 3f shows a closeup of the resulting data at the beginning of the loading process. The hydrogen-induced increase of the spin-flip signal $R^{-+}$ coincides with the increase in resistivity within the statistical error of the count rate (purple box in Fig. 3f). Based on the shift of the resonance peak towards smaller $Q$ during hydrogen loading, one can relate the change of the count rate, $\Delta I$, to the hydrogen concentration:

$$\Delta I = \frac{2h}{w} e^{-1/2} \Delta Q_{res} = \frac{2h}{w} e^{-1/2} \frac{8\pi \bar{N}}{Q_{res}^0} b_H c_H \quad (5)$$

where a Gaussian profile with height $h$ and width $w$ is assumed for the resonance peak, and $\frac{2h}{w} e^{-1/2}$ is the slope of the tangent at

the maximum derivative ($Q^*$) of this profile. We note that expression (5) is valid for small $c_H$, where H-induced changes of the thickness and waveguide enhancement factor can be neglected. As indicated in Fig. 3f, a concentration of ~1 at.% can be detected within a counting time of less than one minute. The constant-$Q$ method is therefore particularly suitable for real-time measurements.

Since Eq. (4) is derived under the assumption of constant film thickness and homogeneous hydrogen distribution, we performed reflectometry simulations to assess the impact of H-induced swelling of the film or H inhomogeneities on the position of the resonant peak (Fig. 3c). This simulation confirms that the variation is negligible at low concentrations, and becomes more important for high H-content ($c_H > 10-15\%$). Therefore, accurate measurements of $c_H$ in kinetic RNR experiments require recalibration based on a full NR profile. To this end, we remeasured the full reflectivity profile at the end of the hydrogenation process in sample 2 (Fig. 2b). Comparison to model calculations confirmed a homogeneous H distribution in both Nb sublayers, as well as the expected thickness increase of the Nb layer of ~10% in the fully loaded state (Inset of Fig. 2b). These data can be used to correct the $c_H$ obtained by RNR. Specifically, Eq. (4) applied to the shift shown in Fig. 3d yields $c_H = 93$ at.%, compared to $c_H = 85$ at.% if the thickness change is taken into account.

In comparison to prior reports of hydrogen absorption in Nb at comparable pressures[21,28], we have observed higher $c_H$ values, and the absorption process was found to be irreversible upon release of the gas pressure. However, the previous studies were performed at higher temperatures (185–200 °C) where the phase diagram shows an equilibrium between $\alpha$ and $\alpha'$ solid solutions. Our experiment on the other hand was performed at room temperature, where formation of the highly hydrogenated $\beta$ phase (niobium hydride) is expected[48] from the bulk phase diagram. The irreversibility of the absorption process even after several months of air exposure also points to the formation of a hydride phase. Definitive confirmation of this phase in our thin films would require further investigations, which go beyond the scope of this study.

## Discussion

The resonant enhancement we have demonstrated is possible for a wide range of materials that can be grown in the form of films with thicknesses between 10 and 100 nm. It requires either a substrate or a capping layer with substantial SLD contrast to the material of interest ($|\Delta \rho| \geq 0.5 \times 10^{-6} Å^{-2}$). In view of the large variation of neutron scattering lengths among chemically similar elements, we do not expect this requirement to impose strong constraints on the method's applicability. An additional requirement is a secondary channel for the detection of the waveguide resonance. We have shown that magnetic neutron scattering can serve as a powerful indicator of the local neutron density. Alternatively, the emission of $\gamma$ or $\alpha$ particles from neutron absorbers such as B, Gd, Cd, and a number of lanthanides can serve as detection channel; (an example is shown in Supplementary Fig. 5). If magnetic or absorbing atoms are not present in the material of interest (as in the case of Nb presented here), additional magnetic or absorbing label layers can be deposited on the substrate, in the middle of the active layer, or on top of it—either in situ or ex situ (see Fig. S5b). As we have done in the present study, the potential impact of such layers on the structure and hydrogen uptake of the active layer has to be carefully considered[49]. An additional possibility for the detection of the waveguide peak is the transmission of neutrons for thin films deposited on low-SLD substrates, such as Si. In this case the

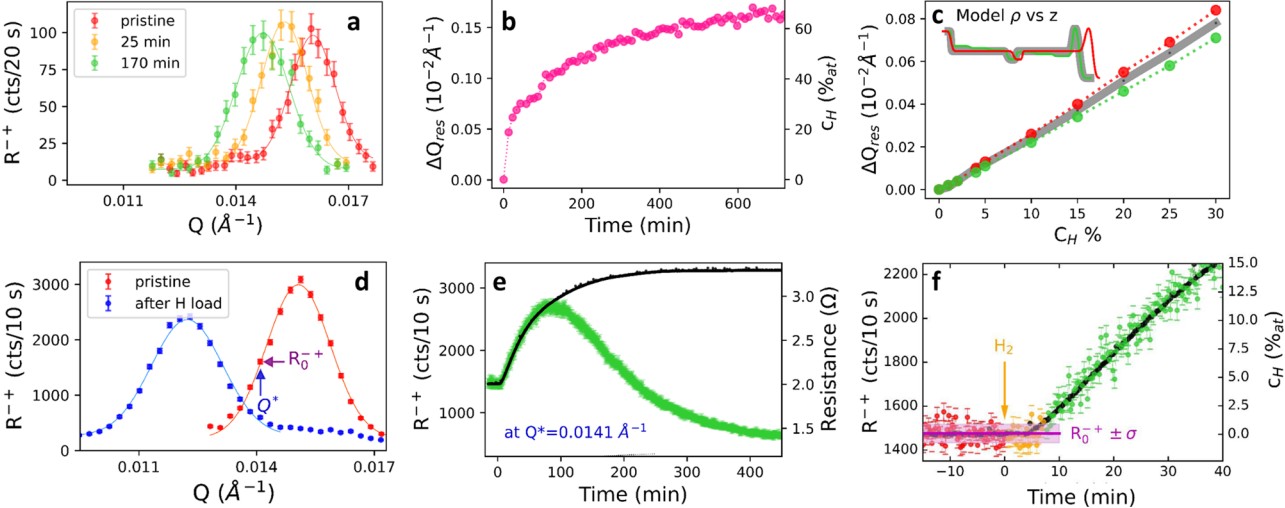

**Fig. 3 Resonant neutron reflectivity during hydrogen loading. a** Typical peak profiles in $Q$-scan mode on sample 1 at the V6 reflectometer. **b** Time dependent shift (left scale) and corresponding hydrogen concentration (right scale) of the resonance peak shown in panel **a**. **c** Simulated shift of the resonance peak for homogeneous H distribution along the film depth and no thickness changes (gray), for homogeneous distribution with 10% thickness change (red), and concentration gradient of H along the layer depth (green). Inset: SLD profiles of different models used for the simulations in panel **c**. **d** Resonance peak measured before and after incorporation of 85 at.% of H on sample 2 at the NREX reflectometer. The lines are the results of fits to Gaussian profiles. The wavevector $Q^*$ used for the fixed-$Q$ mode shown in panel **e** and f is indicated. **e** Time evolution of the intensity at $Q^*$ (green points), and electrical resistance (black points) during H absorption. **f** Closeup of the fixed-$Q$ data in panel **e**, before $H_2$ gas was injected into the vacuum chamber (red data points), right after injection (yellow points), and after an H-induced change of the reflectivity was detected outside the standard deviation $\sigma$ (green points). Resistance data points are shown for comparison in black. Error bars indicate the statistical error.

neutrons are highly transmitted at $Q_{res}$, with a corresponding dip in the conventional NR. If none of these options are practical, the incoherent scattering from H itself can be used as secondary channel, albeit with lower sensitivity[36].

RNR experiments can be performed in two complementary modes: fixed-$Q$ and $Q$-scan. Since it only requires measurements at a single-$Q$-point, the fixed-$Q$ method provides the highest sensitivity to small time-variable H concentrations in in-situ experiments. It is instructive to compare fixed-$Q$ experiments performed with RNR and NR. Based on elementary statistics, the minimum measurement time $t_{min}$ to detect a difference between pristine and H-doped samples can be estimated as follows:

$$t_{min} = \frac{4R}{(R - R_H)^2 * I_0} \quad (6)$$

where $R$ and $R_H$ are the reflectivities of the pristine and hydrogenated samples, respectively, and $I_0$ is the intensity of the incoming neutron beam. Figure 4a, b shows simulated $R$ and $R_H$ for $c_H = 5\%$ in nonpolarized NR and in RNR experiments, respectively, on the structure discussed above with instrumental resolution parameters pertaining to our experiments (Supplementary Table 1). The dashed lines show the calculated $t_{min}$ for both cases. In this calculation we used an incident intensity of the polarized beam $I_0 = 500$ counts per second (as obtained in our experiment at NREX), and took into account that $I_0$ for a nonpolarized beam can be about three times higher. The calculations show that the measurement time for RNR is at least one order of magnitude shorter than for NR ($t_{min} = 0.06$ vs. 2.5 s, respectively).

To explore the universality of the resonant enhancement, we have also considered different systems. Specifically, simulations of NR from a thin Nb film on a Si substrate (a non-waveguide structure) show a $t_{min}$ of 0.25 s, as shown in Fig. 4c. The simulations also demonstrate an additional reduction of $t_{min}$ by a factor of six, if a waveguide resonance is generated by including a thin Pt layer between the substrate and the Nb overlayer (Fig. 4d). Here the secondary channel at play is the transmission of neutrons through the low-SLD substrate, which is maximal around

$Q_{res}$. In this case it is possible to follow the peak in the transmission signal or, as we have shown in Fig. 4d, the corresponding dip in reflectivity. This calculation shows that, independently of its specific implementation, RNR offers an additional substantial enhancement, on top of the sensitivity improvements one can realize by changing the substrate. This advantage is of particular interest for real-time experiments on hydrogen storage materials, where large quantities of H must be measured upon loading and releasing in short times, and on quantum materials such as iron-based superconducors, where even small H concentrations substantially influence the superconducting properties[7].

As single-$Q$ measurements are simply related to the H concentration only over a limited range of concentrations (Fig. 3e), supplementary $Q$-scans of the resonance peak can be carried out periodically in the course of a real-time experiment to guide readjustments of the $Q$ position (Fig. 3a, d). Such scans yield the absolute H concentration in a model-free manner (Eq. (4)) and require much less time than a full NR profile. For the largest concentrations, however, additional NR scans over a wider $Q$-range are desirable to assess H-induced thickness changes and possible concentration gradients (Fig. 2b). All three neutron-based methods are therefore highly complementary.

Neutron methods, in turn, are complementary to nuclear methods such as Nuclear Reaction Analysis (NRA)[50,51] and Elastic Recoil Detection Analysis (ERDA)[52,53], which yield depth profiles of the hydrogen concentration with higher sensitivity, but are affected by radiation damage both due to the structural impact of the high-energy ions and due to charge accumulation inside the material. Since these effects influence the retention of hydrogen during the measurement[32,54], they are particularly problematic for in-situ studies. Neutron-based methods, in contrast, do not have the drawback of radiation damage and allow in-situ loading experiments in $H_2$ gas atmosphere. Table 1 summarizes the characteristics of the most common techniques for H-detection in thin films.

Among a suite of options to enhance the hydrogen sensitivity of neutron reflectometry, the waveguide enhancement we have

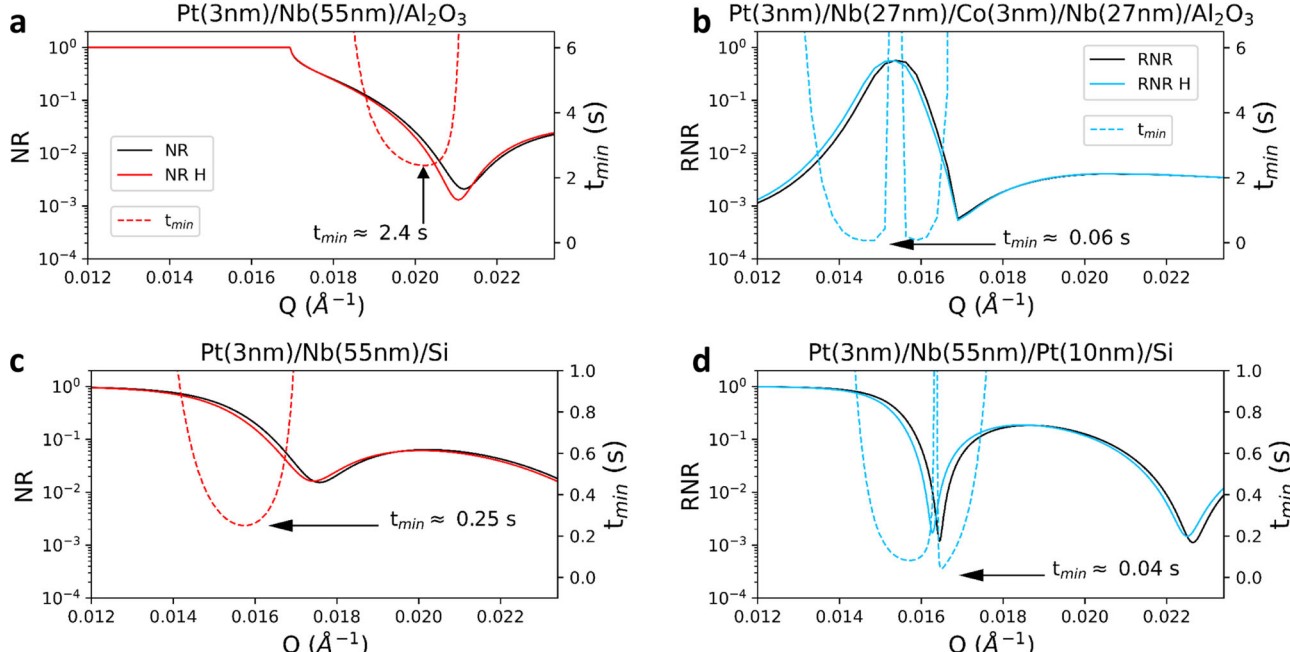

**Fig. 4 Hydrogen sensitivity of NR and RNR in different systems. a** Simulated NR curves for the Pt-capped Nb film on sapphire before (black) and after (red) loading with 5 % hydrogen, with SLDs of 3.91 (3.81) × $10^{-6}$Å$^{-2}$ for pure (5% H-loaded) Nb, respectively. **b** Simulated SF reflectivity at resonance for the same system (with an additional thin Co layer) before (black) and after (blue) H-loading. **c** Simulated NR reflectivity curves for the same film on a silicon substrate. **d** Simulated RNR curves of a Nb film on silicon, with an additional Pt layer at the substrate interface that generates a waveguide structure, before (black) and after (blue) loading with 5% H. The minimum measurement times ($t_{min}$) to detect 5% H are shown as dashed red (blue) lines in conventional NR (RNR).

**Table 1 Comparison between methods for direct detection of hydrogen in thin films.**

|  | Nuclear methods (NRA, ERDA) | Standard neutron reflectometry | Resonant neutron reflectometry |
|---|---|---|---|
| In situ absorption | No | Yes | Yes |
| Simultaneous probe of d | Yes | Yes | No |
| Radiation damage | Yes | No | No |
| Sensitivity limit | ≲0.1 at.% | ~5 at.%[a] | ~1 at.% |
| Measurable kinetics | No | Several minutes | ≲1 min |

[a]Under favorable conditions, ~1% sensitivity can also be achieved by standard reflectometry[23]

introduced offers a particularly versatile tool for in-situ determination of correlations between the absolute hydrogen concentration (as low as one at.%) and electronic properties such as magnetism, metal-insulator transitions, and superconductivity in thin-film structures. The short measurement times that can be realized in this way greatly facilitate real-time studies of the hydrogenation process, thus opening a new avenue for the microscopic understanding of the impact of hydrogen on the electronic structure of solids and solid-state devices.

## Methods

**Sample synthesis and characterization.** Several samples of composition Nb(25 nm)/Co(3 nm)/Nb(25 nm)/Pt(3 nm) were simultaneously grown on Al$_2$O$_3$ substrates (with 1$\bar{1}$02 orientation) of area 10 × 10 mm$^2$ with the molecular beam epitaxy setup of the Jülich Center for Neutron Science at Heinz Maier-Leibnitz Zentrum, following previous work[55]. The rates of deposition were 0.4 Å/s for niobium, 0.1 Å/s for cobalt and 0.2 Å/s for platinum. The substrate temperature was kept at 303 K in order to preserve the bulk-like magnetic properties of the Co layer by minimizing diffusion of cobalt atoms into the niobium layer.

The as-prepared samples were characterized by X-ray diffraction (XRD), SQUID magnetometry, and transport measurements (see Supplementary Fig. 1). XRD was performed in Bragg-Brentano geometry on a diffractometer equipped with a Cu anode as X-ray source and a DECTRIS line detector. The diffraction patterns indicate a polycrystalline texture of the niobium film, with preferential orientation along the (110) plane. Transport measurements using a standard four-

probe device[55] revealed a sharp superconducting transition at ~7 K, slightly lower than in the bulk due to either the proximity of the Co layer[56] or to the reduced dimensionality. Hysteresis loops obtained by SQUID magnetometry at room temperature showed a saturation magnetic moment of 0.96 ± 0.02 kG, consistent with the one obtained by PNR fitting of 0.99 ± 0.05 kG and corresponding to 67% of the bulk value of 1.5 kG.

X-ray reflectometry measurements (with photon wavelength 1.54 Å) were performed at the NREX reflectometer at room temperature (see Supplementary Fig. 2c). The X-ray data were fitted to a model based on the variable thickness and roughness of parameters, using the GenX software[57]. The same model was applied to the neutron reflectometry data discussed below, showing good agreement of the fitted parameters. All relevant parameters retrieved from the fitting of our reflectometric data on pristine and hydrogenated samples are reported in Supplementary Fig. 3.

**Neutron reflectometry.** The polarized neutron reflectometry measurements were conducted at the angle-dispersive NREX reflectometer at the research reactor FRM-II in Garching, Germany[58], with neutrons of wavelength 4.28 Å. Preliminary measurements were carried out at the V6 reflectometer at the BER-II reactor in Berlin, Germany[59,60], with neutron wavelength 4.66 Å. In both cases, the incoming neutrons were polarized before the sample, and transmitted through spin flippers before and after the sample. A polarization analyzer was used to discriminate between spin-flip (SF) and non-spin-flip (NSF) scattering events at the sample. The analyzer transmits the NSF and reflects the SF neutrons into two different channels of the detector (see Supplementary Fig. 1e). All samples were magnetized prior to the neutron experiments with an in-plane magnetic field sufficient to saturate the Co magnetic moment (1.7 kOe at V6 and 4.5 kOe at NREX), and subsequently

mounted in a sealed chamber. The pressure inside the chamber could be tuned by needle valves connected to an Ar(98%)/$H_2$(2%) mixture bottle and turbomolecular pumps. A constant gas pressure of 8 mbar was used, corresponding to 0.16 mbar of pure $H_2$ gas. During the experiments, a magnetic guide field of around 5 and 30 Oe was applied along sample edge at NREX and V6, respectively. The guiding field was chosen to be strong enough to mantain the neutron polarization, and low enough to allow the magnetization to lay on the easy axis direction, along the diagonal of the sample. Following ref. [36], this ensures a constant noncollinear magnetization (the angle between applied field and magnetization $\alpha$ was ~50°) throughout our measurements. Every sample was subjected to H-loading over one single cycle, to ensure the same initial state for every experiment.

## Data availability

The authors declare that all the data supporting the findings of this study are either shown in the main and supplementary text or available from the corresponding author upon request.

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

## Acknowledgements
Among the experiments shown here, some were among the last conducted at the V6 reflectometer of the BER-II reactor. We would like to thank HZB for the allocation of neutron time at V6, and the whole staff of the reactor, especially Roland Steitz, Heike Gast, Dirk Wallacher, and Nico Grimm, for their help and hospitality during this and all previous experiments. We would also like to thank F. Klose and C. Rehm for useful discussions. We acknowledge financial support of German Research Foundation (Deutsche Forschungsgemeinschaft, DFG, Project No. 107745057–TRR80).

## Author contributions
L.G., Y.K., and B.K. conceived the study. S.P. prepared the samples. L.G. performed the preliminary characterization of the samples. L.G., Y.K., L.S., and M.A.P. performed the PNR experiments at V6. L.G., Y.K., and T.K. carried out the PNR measurements at NREX. L.G. performed the reflectivity simulations and data analysis. All authors discussed the experimental results. L.G. and Y.K. wrote the manuscript with substantial input from T.K. and B.K., and with contributions from all authors.

## Funding

## Competing interests
The authors declare no competing interests.
