## [Peer Review File · Nature Communications]

REVIEWER COMMENTS

Reviewer #1 (Remarks to the Author):

Guasco et al. present a new neutron-based method to determine the absolute hydrogen concentration in thin films. Their method is novel and can be useful in some studies where hydrogen needs to be quantified in thin films and would as such be worth publishing in a well-respected journal such as Nature Communications. Although the method may certainly have advantages over other conventional methods to quantify the amount of hydrogen, the authors exaggerate the advantages of their method and the disadvantages of established techniques and do not compare their method to established methods in an appropriate context. Furthermore, I think the manuscript would benefit if neutron scattering jargon is either avoided or explained as this would make the manuscript more appealing to a broader range of scientists. I advise that this manuscript can be considered for Nature Communications after a major revision considering all points below. Especially, the authors should compare their technique in a fair and transparent way to other techniques presently used to quantify hydrogen amounts, stating not only its advantages of their developed technique but also the disadvantages and limitations (need for polarized neutrons, thickness of film, flat surfaces, homogeneity of the sample etc.), and not exaggerate the disadvantages or downplay the performance of other techniques. In addition, the manuscript should include a verification of the measured hydrogen content using the new technique with e.g. specular NR. One can then also use this verification experiment as a basis to compare the advantages/disadvantages of the proposed method to existing methods.

Detailed comments

The authors often write 'neutron reflectometry' in their manuscript to refer to resonance (enhanced) neutron reflectometry. This is sometimes misleading as it is sometimes used to refer to the authors new method or to conventional specular neutron reflectometry.

The authors write: 'A thin Co layer inserted into the Nb film serves as an indicator of the local neutron density via spin-flip scattering and thus allows tracing of the resonance position'. Can the authors explicitly state whether such indicator layers are always required? Would it be possible to use a magnetic substrate instead or insert the Co layer on top of the substrate? If an additional indicator layer is required, this would be a major downside of the technique as this is not required in specular NR and alters the studied system significantly. In this case, if one wants to compare the performance of the newly reported technique to specular NR, one should compare with a sample that does not have an additional Co layer but simply a sample with a Nb and a capping layer.

The authors write: 'Its thickness of 3 nm is small enough to allow fast hydrogen transmission, yet large enough to generate a magnetization close to the bulk value (see Methods).' Cobalt is not

known for a high diffusivity of hydrogen, and despite it being relatively thin, it may form a substantial barrier for the hydrogenation of the Nb layer underneath. Could it be the case that first the top Nb layer hydrogenates, and then the bottom one? Can the measurements discriminate between this scenario and homogenous hydrogenation?

Why don't the authors share the data of 'additional neutron reflectivity and complementary x-ray reflectivity data' referred to on page 2 in a supplement?

The authors write: 'The retrieved nuclear and magnetic SLD profiles, shown in the inset of Fig. 2, indicate an excellent match between nominal and measured thicknesses and SLDs for all layers and interfacial roughnesses well below the layer thickness.' This statement is hard to be checked based on the small inset in Fig 2. Can the authors provide a table with the fitted parameters based on all considered techniques? In addition, I presume this has been measured on the as-prepared sample, i.e. before hydrogenation.

The manuscript is quite technical for a general journal as Nature Communications, requiring knowledge about neutron scattering. For example, statements including 'Condition (3a) specifies that resonances occur in the vicinity of the critical edge for total external reflection.' cannot be followed by a general audience (e.g. material scientist that want to quantify the amount of hydrogen in materials), as to them it is not sure what is meant by 'critical edge' or 'total external reflection' (obviously, for people working with NR this is fine, so for a more specialized journal this would be ok.). Related to this, I would advise the authors to explain their method in a way that can be followed by non-neutron users including why they need polarized neutrons and also schematically illustrate the scattering geometry and the experimental set-up.

The authors write on the top of page 2: 'We consider a generic three-layer potential comprising a capping layer, an active layer, and a substrate (layer index $i = 1 - 3$, respectively).', while Figure 1 (next to it and only referred to on page 3) displays a much more complex situation.

How is equation (4) affected by (hydrogenation-induced) changes in the film thickness?

The title 'Resonant neutron reflectometry for hydrogen detection' is misleading to me. To me, detection is to detect whether something is present or not (i.e. binary). However, the authors claim that they can quantify the amount of hydrogen. Indeed, in the first line of the abstract, the authors talk about 'The detection and quantification of hydrogen'.

Can the authors explain what is meant by 'The incorporation of hydrogen in materials is central

for the vision of a hydrogen economy"?

In the abstract, the authors write: 'Especially in thin films, however, detection of small hydrogen concentrations presents a formidable challenge.' (I do not necessarily agree with this statement), while in the second paragraph of the introduction the authors write: 'Thin films are the basis of most solid-state devices and are particularly suited for quantitative investigations of the influence of hydrogen on different physical properties of materials, because homogeneous profiles of both hydrogen and the quantity of interest can be easily obtained, and measurements can be carried out in well-defined geometries.' To me, these statements seem to be at variance with each other. Can the authors comment on this?

The authors write: 'In practice, however, the small amount of material combined with the limited intensity of neutron beams have restricted the sensitivity of this method to concentrations exceeding ~10 atomic %, which is inadequate for many potential applications.' The authors provide no reference for this statement. To me this statement is highly misleading. For accurate determination of the hydrogen content in thin films, one should not compare with neutron imaging but with specular neutron reflectometry. With this technique, studies have been performed in which the sensitivity was much smaller than ~10 atomic percent, typically about 1%. See, e.g., the following articles for an overview of previous NR studies:

- Fritzsche, H., Huot, J., & Fruchart, D. (Eds.). (2016). Neutron scattering and other nuclear techniques for hydrogen in materials. Switzerland: Springer International Publishing.
- Dura, J. A., Rus, E. D., Kienzle, P., & Maranville, B. B. (2017). Nanolayer analysis by neutron reflectometry.
- Bannenberg, L. J., Heere, M., Benzidi, H., Montero, J., Dematteis, E. M., Suwarno, S., ... & El Kharbachi, A. (2020). Metal (boro-) hydrides for high energy density storage and relevant emerging technologies. International Journal of Hydrogen Energy.

As such, my advice would be to provide a fair comparison, stating that most studies use now specular neutron reflectometry to determine the hydrogen content in a thin film (rather than imaging) and then introduce the proposed alternative method.

Related to the above, the authors write: 'We have taken advantage of waveguide resonances resulting from the formation of neutron standing waves in thin films [21] to increase the sensitivity by at least one order of magnitude.' Again, this is a false comparison and thus a misleading statement. Please adapt accordingly.

Related to the above: 'Notably, resonant neutron reflectometry yields absolute measurements of the hydrogen concentration (in contrast to x-ray reflectometry, which monitors hydrogen incorporation indirectly via its impact on the layer thickness [22{24}]): why do the authors compare to X-ray reflectometry here and not to specular NR?.'

The authors state: 'In particular, higher sensitivity is typically required for in situ studies that yield parametric correlations between the hydrogen content and the physical properties of interest and offer insight into the kinetics of hydrogen incorporation in real time.' Such studies have also been performed with specular neutron reflectometry (e.g. correlating optical transmission with hydrogen content, as e.g. done in the field of metal hydride hydrogen sensors) or studying the hydrogen sorption kinetics of metal hydride thin films.

The authors write: 'Our experiment was performed at room temperature, where the formation of highly hydrogenated phase (niobium hydride) is expected'. Comment: the authors refer to bulk phase diagrams. In the case of thin films, phase transitions may be suppressed (as is e.g. the case in Ta films). Also, there is plenty of literature concerning Nb films by e.g. A. Pundt showing nanoconfinement effects.

In the discussion, the authors write: 'Moreover, the substantially shorter measurement times [w.r.t. specular NR] (minutes versus hours with typical instruments and sample geometries) allow realtime studies of the incorporation process.' I find this comparison misleading. Kinetic studies of the hydrogenation of thin films have been performed with minutes resolution, even on NR beamlines with relatively low fluxes as e.g. Offspec at ISIS on e.g. magnesium, palladium-based, hafnium and tantalum thin films.

The authors write: 'Calibration measurements over a wider Q-range are required for large c_H , where H-induced modification of the thickness of the active layer may become relevant, or in cases where strong concentration gradients are present.' Can the authors comment on the effect of changing thickness on the reliability of their determination of c_H ? Typically, metal hydride films expand significantly during hydrogenation (about 10-20% per Hydrogen atom inserted). An advantage of Specular NR is in this case that also the thickness is probed simultaneously.

Related to the above, Fig. 1b is not realistic as hydrogenation of the Nb layer to $NbH_{0.85}$ would induce a significant expansion of the Nb layer thickness by about 10%.

Have the authors verified the determined CH in Fig. 3 with results obtained from specular NR? What is the correspondence?

Can the authors include a fair comparison, including not only the advantages but also the disadvantages of their technique with other techniques to determine hydrogen amounts in thin films? This can be in table form and can e.g. include specular NR, NRA etc.

The authors state that no radiation damage is a major advantage of neutron techniques to quantify hydrogen as opposed to 'nuclear methods'. Another advantage is that neutron based techniques can be performed in situ, e.g. under a hydrogen atmosphere, while nuclear methods often require vacuum.

In their analysis of Fig. 3, the authors not only use the peak position of the resonance but also its intensity to determine the hydrogen content. They write: 'Eq. (4) again allows conversion into the absolute H concentration.'. How can Eq. 4, in which intensity is not a parameter, be used to convert measured intensity into a hydrogen concentration?

Related to the above, there only seems to be a correlation between CH and the measured intensity for low CH (Fig. 3c). Can the authors discuss the limitations of this fixed-Q technique? To me it seems that this would make the fixed-Q method unsuitable for almost any application.

Reviewer #2 (Remarks to the Author):

The authors use their knowledge on neutron reflectivity to develop methodology for determining hydrogen concentration in thin layers. The neutron scattering part is well thought of and convincing with respect to both the experiments as well as the analysis.

The perceived weakness of the manuscript is twofold:

(1) Convincing arguments for how neutron reflectivity will enable "hydrogen economy" are not presented in the manuscript. The motivation of the work can therefore be strengthened. There are two ways around this: Chose alternative motivation for the work or provide compelling arguments

for why this technique is as important as indicated in the beginning of the manuscript for the hydrogen economy.

(2) The discussion on the sample structure and the consequences of the choices made needs to be strengthened to grant full appreciation of the results. Some of those are listed below.

2.a) Information on the crystalline structure of the absorbing layers is needed. Is there a difference in the crystalline quality/texture of the Nb layers (below/above the Co layer)?

2.b) By adding Co layer midway in the Nb stack, makes two layers. These can have completely different properties with respect to hydrogen uptake at low concentrations. The effects at higher concentrations are expected to be smaller. See e.g. Emil Johansson et al 2004 J. Phys.: Condens. Matter 16 1165 for extended discussion on the impact of interfaces on the hydrogen uptake. This aspect needs to be taken into account if advocating the approach as a generic technique for determining the hydrogen concentration in materials.

2.c) Hydrogen embrittlement can be profound, even in thin layers. Polycrystalline materials can display large changes upon hydrogen cycling as compared to single crystalline materials, depending on the details of the hydrogen induced volume changes of the material. This needs to be dwelled upon for completeness. It is also worth to highlight the difference in the clamping of the strain field, depending on the crystallographic relation of the substrate and the film.

2.d) Having magnetic layer in the centre allows for a change in contrast by changing the magnetic direction of that layer. For example, the magnetic direction of the Co layer (with respect to the polarisation axis of the neutrons) can be 0, $\pi/2$, and π . This gives 3 different conditions that can be used to establish better information on the scattering profile and thereby the concentration and the concentration profile within the sample. This can be used to strengthen the communication.

The examples provided above are not complete. Furthermore, depending on which direction the authors decide to take (see under (1) above), different weights need to be given to the issues raised.

Reviewer #3 (Remarks to the Author):

This is a very well thought out experiment and superbly executed. Physics of the idea is well established and as the authors correctly point out, neutron standing waves provide substantial field enhancement in well fabricated fabricating quantum resonators.

This allows one to study thin films by neutron reflectivity near the critical angle and study response from buried layers by exploiting the emission of gamma ray, incoherent scattering, in-plane diffraction and in this article, by looking at the spin flip channel of magnetic scattering in thin films.

The samples made by MBE are very well characterized by various methods. And the authors convincingly demonstrate that Q shift in the resonance of spin flip channel shows very good sensitivity to low levels of hydrogen loading in buried Niobium films.

I have only positive comments to make about this article except for the following.

As I mentioned above, these MBE samples are nearly ideally made. Roughness between the films can severely degrade the sharpness of the peaks observed in reflectivity measurements.

At least for the detection of low end of the hydrogen loading quite small subtle shift in Q of resonance peak is required. So it is not clear clear to me how generally applicable is this broad statement in the abstract "Resonance-enhanced neutron reflectometry thus allows direct, non-destructive measurements of the hydrogen concentration in thin-film structures,

with sensitivity high enough for real-time in-situ studies"

The paper it self is quite well written.

The only small thing I would point out is that in Fig.3 CAPTION it would be helpful to actually point out which parts (a,b,c,d)

belong to which of the two samples that were studied (even though the authors do state that in the text of the article).

REVIEWER COMMENTS

Reviewer #1 (Remarks to the Author):

Guasco et al. present a new neutron-based method to determine the absolute hydrogen concentration in thin films. Their method is novel and can be useful in some studies where hydrogen needs to be quantified in thin films and would as such be worth publishing in a well-respected journal such as Nature Communications. Although the method may certainly have advantages over other conventional methods to quantify the amount of hydrogen, the authors exaggerate the advantages of their method and the disadvantages of established techniques and do not compare their method to established methods in an appropriate context. Furthermore, I think the manuscript would benefit if neutron scattering jargon is either avoided or explained as this would make the manuscript more appealing to a broader range of scientists. I advise that this manuscript can be considered for Nature Communications after a major revision considering all points below. Especially, the authors should compare their technique in a fair and transparent way to other techniques presently used to quantify hydrogen amounts, stating not only its advantages of their developed technique but also the disadvantages and limitations (need for polarized neutrons, thickness of film, flat surfaces, homogeneity of the sample etc.), and not exaggerate the disadvantages or downplay the performance of other techniques. In addition, the manuscript should include a verification of the measured hydrogen content using the new technique with e.g. specular NR. One can then also use this verification experiment as a basis to compare the advantages/disadvantages of the proposed method to existing methods.

We would like to thank the referee for carefully reading our manuscript and for her/his insightful comments. We have rewritten the text according to these comments, especially with regard to the comparison with other techniques for H detection and profiling. We have included the full NR profile of the hydrogenated sample with $c_H=85\%$ together with a corresponding fit (Fig. 2b), which shows that the entire Nb is hydrogenated homogeneously and allows an assessment of H-induced swelling and its impact on the position of the waveguide resonance peak. In the discussion, we added a quantitative comparison of the suggested method with standard NR. We also discuss drawbacks of our method, and we added a table with comparisons to other techniques (Table 1). Finally, we also improved the description of neutron reflectometry in language understandable for non-experts.

1. The authors often write 'neutron reflectometry' in their manuscript to refer to resonance (enhanced) neutron reflectometry. This is sometimes misleading as it is sometimes used to refer to the authors new method or to conventional specular neutron reflectometry.

We have clarified this distinction by introducing two acronyms (NR and RNR) for standard and resonance-enhanced reflectivity, respectively.

2. The authors write: 'A thin Co layer inserted into the Nb film serves as an indicator of the local neutron density via spin-flip scattering and thus allows tracing of the resonance position'. Can the authors explicitly state whether such indicator layers are always required? Would it be possible to use a magnetic substrate instead or insert the Co layer on top of the substrate? If an additional indicator layer is required, this would be a major downside of the technique as this is not required in specular NR and alters the studied system significantly. In this case, if one wants to compare the performance of the newly reported technique to specular NR, one should compare with a sample that does not have an additional Co layer but simply a sample with a Nb and a capping layer.

A discussion of possible ways to detect neutron resonances was already included in the previous version of the manuscript (in the sections "Neutron waveguide resonances" and "Discussion"). The answer to the reviewer's query is that magnetic layers are not always required. Alternatively, one can use e.g. γ or α particle detection channels or incoherent scattering. The latter methods do not need polarization analysis, but a dedicated detection device is required. For proof-of-principle demonstration, we found that a magnetic layer in the center of the waveguide is most instructive. The layer is so thin that the system is not significantly altered. In particular, the layer does not significantly affect the diffusion of hydrogen into the film, as demonstrated directly by the full NR profile at the end of the run. However, such a layer can be also placed directly on the substrate or on top of the active layer. We added further discussion of this issue at the end of the section "Neutron waveguide resonances", and address this issue prominently at the beginning of the "Discussion" section.

3. The authors write: 'Its thickness of 3 nm is small enough to allow fast hydrogen transmission, yet large enough to generate a magnetization close to the bulk value (see Methods).' Cobalt is not known for a high diffusivity of hydrogen, and despite it being relatively thin, it may form a substantial barrier for the hydrogenation of the Nb layer underneath. Could it be the case that first the top Nb layer hydrogenates, and then the bottom one? Can the measurements discriminate between this scenario and homogenous hydrogenation?

The H-transmission properties of such a thin Co layer have not been studied directly, to the best of our knowledge. However, several conventional NR experiments on multilayers comprising comparably thick layers of Fe (whose solubility for hydrogen is similar to that of Co) have demonstrated homogeneous hydrogenation under conditions similar to ours [10,11,22]. Prompted by the referee's question, we have performed an analysis of the full NR profile of our hydrogenated sample, which directly demonstrates closely similar concentrations of hydrogen in both Nb sublayers above and below the Co layer. We included this result in Fig. 2 and added the following sentences to the "Sample design..." section: "As magnetic label layer we used a 3 nm thick Co layer inserted in the middle of the Nb layer (Fig. 1a). The choice of material, thickness, and position of the label layer was dictated by the following considerations. The layer is thin enough not to distort the waveguide properties of the system (Fig. 1c), but thick enough to be ferromagnetic with a moment approaching the bulk value (Suppl. Fig. S1d). We also expect that the thin Co layer, being similar to the Fe layers used in other hydrogenation

experiments [10, 11, 22], has a high transmittance for hydrogen atoms; this was directly confirmed in our experiments (see below). In contrast to Fe, however, Co is known to form sharp interfaces with Nb [43,44]. Finally, we placed the label layer in the middle of the structure for the sake of proof-of-principle demonstration. Alternatively, one may use a label layer on top of the active layer or on the bottom of the substrate, as demonstrated in Ref. [33].”

4. Why don't the authors share the data of 'additional neutron reflectivity and complementary x-ray reflectivity data' referred to on page 2 in a supplement?

In the newly added Supplementary Material, we include NR, x-ray reflectivity, magnetization, and transport data as well as a schematic of the PNR experiment.

5. The authors write: 'The retrieved nuclear and magnetic SLD profiles, shown in the inset of Fig. 2, indicate an excellent match between nominal and measured thicknesses and SLDs for all layers and interfacial roughnesses well below the layer thickness.' This statement is hard to be checked based on the small inset in Fig 2. Can the authors provide a table with the fitted parameters based on all considered techniques?

In the original version of the manuscript, the thickness and roughness values extracted from fits to the reflectometric profiles were already reported in the Methods section. In the new version, we included an additional table (Table S1) in the Supplementary Material.

6. In addition, I presume this has been measured on the as-prepared sample, i.e. before hydrogenation.

Indeed, the NR was measured on a pristine sample, as explained in the caption of Fig. 2 and in the main text. In the new version, we also included NR data for the hydrogenated sample (Fig. 2b).

7. The manuscript is quite technical for a general journal as Nature Communications, requiring knowledge about neutron scattering. For example, statements including 'Condition (3a) specifies that resonances occur in the vicinity of the critical edge for total external reflection.' cannot be followed by a general audience (e.g. material scientist that want to quantify the amount of hydrogen in materials), as to them it is not sure what is meant by 'critical edge' or 'total external reflection' (obviously, for people working with NR this is fine, so for a more specialized journal this would be ok.). Related to this, I would advise the authors to explain their method in a way that can be followed by non-neutron users including why they need polarized neutrons and also schematically illustrate the scattering geometry and the experimental set-up.

Fig. 1a presents a simple scheme explaining reflectometric experiments (with pictorial representations of the incoming angle θ and wavevector transfer Q). In addition, we now included a more detailed explanation of the polarized neutron reflectometry setup in the Supplementary Material (Fig. S1e). An explanation of the term "total reflection" was added to the section "Neutron waveguide resonances", and the term "critical edge" is no longer used.

8. The authors write on the top of page 2: 'We consider a generic three-layer potential comprising a capping layer, an active layer, and a substrate (layer index $i = 1 - 3$, respectively).', while Figure 1 (next to it and only referred to on page 3) displays a much more complex situation.

We modified Fig. 1a to show which layers are labelled by the indices $i=1,2$ and 3. We added a sentence to explain that the Co layer in the middle of the active layer is thin enough in order not to disturb the resonance properties, and also show this schematically in Fig. 1a.

9. How is equation (4) affected by (hydrogenation-induced) changes in the film thickness?

Eq. 4 is derived under the assumption of constant thickness d_2 , as explained right above the formula. We also added calculations of ΔQ_{res} for the cases of swelling thickness and nonhomogeneous $c_H(z)$ (see Fig. 3c and associated text). These calculations show that up to $c_H \sim 10\text{-}15\%$ the shift of the resonance is nearly unaffected by thickness variations. For higher concentrations, the error increases and amounts to 8% at full hydrogenation. Accurate measurements for large c_H therefore require an additional calibration measurement at the end of the run. This is now explained in the text.

10. The title 'Resonant neutron reflectometry for hydrogen detection' is misleading to me. To me, detection is to detect whether something is present or not (i.e. binary). However, the authors claim that they can quantify the amount of hydrogen. Indeed, in the first line of the abstract, the authors talk about 'The detection and quantification of hydrogen'.

We agree with the referee that "detection" might be understood in this way. In our view, however, this term is also commonly used for quantitative measurements. For example, a neutron detector measures not only the presence/absence of neutrons but also provides quantitative information on the neutron flux. We therefore prefer to keep the original title for brevity. However, if the referee and the editors insist, we are ready to adopt a different title such as 'Resonant neutron reflectometry for hydrogen detection and quantification'.

11. Can the authors explain what is meant by 'The incorporation of hydrogen in materials is central for the vision of a hydrogen economy'?

We reformulated this sentence: "In most of the technologies underlying the vision of a 'hydrogen economy', hydrogen is present either within or in immediate proximity to materials. Sensitive methods to quantify the incorporation of hydrogen in solids are hence of intense interest across a wide variety of research fields..." After laying out the general context, we describe the specific roles of neutron reflectometry and of our resonance enhancement method later in the introductory paragraphs.

12. In the abstract, the authors write: 'Especially in thin films, however, detection of small hydrogen concentrations presents a formidable challenge.' (I do not necessarily agree with this statement), while in the second paragraph of the introduction the authors write: 'Thin films are the basis of most solid- state devices and are particularly suited for quantitative investigations of the influence of hydrogen on different physical properties of materials, because homogeneous profiles of both hydrogen and the quantity of interest can be easily obtained, and measurements can be carried out in well-defined geometries.' To me, these statements seem to be at variance with each other. Can the authors comment on this?

We agree that these statements can be understood as contradictory. We have therefore rephrased the statement in the abstract: “However, establishing quantitative correlations between the hydrogen content and the physical properties of solids remains a formidable challenge.”

13. The authors write: ‘In practice, however, the small amount of material combined with the limited intensity of neutron beams have restricted the sensitivity of this method to concentrations exceeding ~10 atomic %, which is inadequate for many potential applications.’ The authors provide no reference for this statement. To me this statement is highly misleading. For accurate determination of the hydrogen content in thin films, one should not compare with neutron imaging but with specular neutron reflectometry. With this technique, studies have been performed in which the sensitivity was much smaller than ~10 atomic percent, typically about 1%. See, e.g., the following articles for an overview of previous NR studies:

- Fritzsche, H., Huot, J., & Fruchart, D. (Eds.). (2016). Neutron scattering and other nuclear techniques for hydrogen in materials. Switzerland: Springer International Publishing.
- Dura, J. A., Rus, E. D., Kienzle, P., & Maranville, B. B. (2017). Nanolayer analysis by neutron reflectometry.
- Bannenberg, L. J., Heere, M., Benzidi, H., Montero, J., Dematteis, E. M., Suwarno, S., ... & El Kharbachi, A. (2020). Metal (boro-) hydrides for high energy density storage and relevant emerging technologies. International Journal of Hydrogen Energy.

As such, my advice would be to provide a fair comparison, stating that most studies use now specular neutron reflectometry to determine the hydrogen content in a thin film (rather than imaging) and then introduce the proposed alternative method.

We agree with the referee that our method should be compared to NR rather than to neutron imaging. To avoid possible misunderstanding, we rewrote the introduction and added references to pertinent prior NR work. We would like to thank the referee for the provided literature, which we partly included in the new version of the manuscript (see ref. [21]). A search for original articles dealing with several at. % of hydrogen yielded only a single article (H. Fritzsche et al. Int. J. Hydrog. Energy 2012), where $c_H \approx 15$ at.% was measured (see Figs. 3 and 7 in this work, now cited as Ref. 23). Hence, we continue to state that prior experiments with conventional NR have demonstrated hydrogen concentration measurements of the order of 10%. However, if the referee knows about additional references we have missed, we will be sincerely grateful if (s)he provides us with a reference of original work where less than 10 atomic percent of hydrogen was reliably detected using conventional NR. Finally, we point to our new Fig. 4, which was inserted upon request of Reviewer 3, where we directly and quantitatively compare the sensitivities of NR and RNR.

14. Related to the above, the authors write: ‘We have taken advantage of waveguide resonances resulting from the formation of neutron standing waves in thin films [21] to increase the sensitivity by at least one order of magnitude.’ Again, this is a false comparison and thus a misleading statement. Please adapt accordingly.

Despite the considerations laid out under point 13, we agree with the referee that this direct comparison is not quite appropriate, because the full reflectometric profiles measured by conventional NR yield more (and partially complementary) information than the method we are introducing. We have therefore rephrased the sentence to read: “We have taken advantage of waveguide resonances resulting from the formation of neutron standing waves in thin films to substantially increase the

sensitivity of neutron reflectometry for real-time hydrogenation experiments.” We have also partially rewritten the Discussion on page 6 to emphasize the complementarity of resonant and non-resonant neutron reflectometry.

15. Related to the above: ‘Notably, resonant neutron reflectometry yields absolute measurements of the hydrogen concentration (in contrast to x-ray reflectometry, which monitors hydrogen incorporation indirectly via its impact on the layer thickness [22-24]): why do the authors compare to X-ray reflectometry here and not to specular NR?’

We have rephrased this sentence in response to the referee’s comments: “ ... Neutron methods yield absolute measurements of the hydrogen concentration (in contrast to x-ray reflectometry or diffraction ...)”

16. The authors state: ‘In particular, higher sensitivity is typically required for in situ studies that yield parametric correlations between the hydrogen content and the physical properties of interest and offer insight into the kinetics of hydrogen incorporation in real time.’ Such studies have also been performed with specular neutron reflectometry (e.g. correlating optical transmission with hydrogen content, as e.g. done in the field of metal hydride hydrogen sensors) or studying the hydrogen sorption kinetics of metal hydride thin films.

We have cut this sentence in response to the referee’s comment.

17. The authors write: ‘Our experiment was performed at room temperature, where the formation of highly hydrogenated phase (niobium hydride) is expected’. Comment: the authors refer to bulk phase diagrams. In the case of thin films, phase transitions may be suppressed (as is e.g. the case in Ta films). Also, there is plenty of literature concerning Nb films by e.g. A. Pundt showing nanoconfinement effects.

We updated the discussion of hydride formation (see page 5, left column).

18. In the discussion, the authors write: ‘Moreover, the substantially shorter measurement times [w.r.t. specular NR] (minutes versus hours with typical instruments and sample geometries) allow realtime studies of the incorporation process.’ I find this comparison misleading. Kinetic studies of the hydrogenation of thin films have been performed with minutes resolution, even on NR beamlines with relatively low fluxes as e.g. Offspec at ISIS on e.g. magnesium, palladium-based, hafnium and tantalum thin films.

The referee is right, the formulation of the sentence is (unintentionally) misleading. We did not mean to say that kinetics studies are not possible with NR. Prompted by the referee’s remark, we changed the wording “... allow real-time studies ...” to “... facilitate real-time studies ...” Based in part on the information provided by the referee, we also added Refs. [23-25.] Refs. 24 and 25 report kinetics studies at the Offspec spectrometer at ISIS.

19. The authors write: ‘Calibration measurements over a wider Q-range are required for large cH, where H-induced modification of the thickness of the active layer may become relevant, or in cases where strong concentration gradients are present.’ Can the authors comment on the effect of changing

thickness on the reliability of their determination of CH? Typically, metal hydride films expand significantly during hydrogenation (about 10-20% per Hydrogen atom inserted). An advantage of Specular NR is in this case that also the thickness is probed simultaneously.

We agree with the reviewer. We added Fig. 3c to elucidate the effect of thickness changes. We changed the text in the Discussion to clarify that the RNR method is to be considered as a complement of conventional NR. Whereas with NR it is possible to precisely characterize the composition of the pristine and final states, the resonant method provides insight into the kinetics of the H absorption process on a timescale of seconds. Finally, we included NR data for the fully hydrogenated sample (Fig. 2b), together with an analysis which shows that the Nb film did expand as expected. Again, we note that RNR is applicable as a standalone method for small concentrations of hydrogen, and that additional NR measurements are only required for large concentrations.

20. Related to the above, Fig. 1b is not realistic as hydrogenation of the Nb layer to NbH_{0.85} would induce a significant expansion of the Nb layer thickness by about 10%.

The reviewer is correct (as now shown on the real SLD profiles before and after H in the inset of Fig.2b). However, Fig. 1b is a simple model calculation intended to illustrate the principle of the method, namely the effect of H loading on the resonance field, without complicating the picture with secondary effects such as the swelling of the film. We added a related comment to the text and to the caption of Fig. 1.

21. Have the authors verified the determined CH in Fig. 3 with results obtained from specular NR? What is the correspondence?

We included NR data for the fully hydrogenated sample (Fig. 2b) which shows that a 85% concentration of hydrogen was loaded in the end. Without correction, Equation 4 (which does not include the effect of thickness modifications) yields 93%. Prompted by the reviewer's comment, we have added panel c to Fig. 3, which quantifies the correction to Eq. 4 due to H-induced swelling as a function of c_H .

22. Can the authors include a fair comparison, including not only the advantages but also the disadvantages of their technique with other techniques to determine hydrogen amounts in thin films? This can be in table form and can e.g. include specular NR, NRA etc.

We have revised the Discussion of the different neutron methods and their complementarity on page 6, and we added Table 1 that briefly summarizes the strengths and weaknesses of each method.

23. The authors state that no radiation damage is a major advantage of neutron techniques to quantify hydrogen as opposed to 'nuclear methods'. Another advantage is that neutron based techniques can be performed in situ, e.g. under a hydrogen atmosphere, while nuclear methods often require vacuum.

We thank the referee for this comment, which we gladly included in the text on page 1.

24. In their analysis of Fig. 3, the authors not only use the peak position of the resonance but also its intensity to determine the hydrogen content. They write: 'Eq. (4) again allows conversion into the

absolute H concentration.'. How can Eq. 4, in which intensity is not a parameter, be used to convert measured intensity into a hydrogen concentration?

We added Eq. (5) to explain how the amplitude of the resonance peak is related to concentration.

25. Related to the above, there only seems to be a correlation between CH and the measured intensity for low CH (Fig. 3c). Can the authors discuss the limitations of this fixed-Q technique? To me it seems that this would make the fixed-Q method unsuitable for almost any application.

Since it only requires measurements at a single Q-point, the fixed-Q method provides the highest sensitivity to small time-variable H concentrations in in-situ experiments. This is of interest, for instance, for iron-based superconductivity where even small H concentrations have a significant influence on the superconducting properties (Ref. [7]). As a stand-alone method, the single-Q measurement is simply related to the H concentration only for small concentrations (as shown in Fig. 3e). However, if Q-scans are performed at certain time intervals, and Q is adjusted accordingly, real-time experiments can benefit from the single-Q method's sensitivity over extended ranges of concentration. The complementarity of fixed-Q, Q-scan, and standard NR methods is now explicitly discussed on page 6.

Reviewer #2 (Remarks to the Author):

The authors use their knowledge on neutron reflectivity to develop methodology for determining hydrogen concentration in thin layers. The neutron scattering part is well thought of and convincing with respect to both the experiments as well as the analysis.

We would like to thank the referee for the careful reading of the manuscript and for the helpful and constructive comments.

The perceived weakness of the manuscript is twofold:

(1) Convincing arguments for how neutron reflectivity will enable "hydrogen economy" are not presented in the manuscript. The motivation of the work can therefore be strengthened. There are two ways around this: Chose alternative motivation for the work or provide compelling arguments for why this technique is as important as indicated in the beginning of the manuscript for the hydrogen economy.

This comment is similar to the comment #11 of referee 1. In response to both of these comments, we have rewritten the introductory sentence to spell out the general "hydrogen economy" context more clearly. Later in the introductory paragraph, we explain the specific role our method can play in this context, namely understanding and control of the impact of hydrogenation on electronic materials and devices.

(2) The discussion on the sample structure and the consequences of the choices made needs to be strengthened to grant full appreciation of the results. Some of those are listed below.

2.a) Information on the crystalline structure of the absorbing layers is needed. Is there a difference in the crystalline quality/texture of the Nb layers (below/above the Co layer)?

The Supplemental Material attached to the revised version comprises x-ray, neutron, magnetometry, and transport data. The X-ray diffraction data confirmed polycrystalline growth of both Nb layers in the (110) direction, without any noticeable differences in the macro- or microstructure. The presence of a single, sharp superconducting transition for our pristine sample with transition temperature 6.8 K indicates a good quality and homogeneity of both Nb sublayers.

2.b)By adding Co layer midway in the Nb stack, makes two layers. These can have completely different properties with respect to hydrogen uptake at low concentrations. The effects at higher concentrations are expected to be smaller. See e.g. Emil Johansson et al 2004 J. Phys.: Condens. Matter 16 1165 for extended discussion on the impact of interfaces on the hydrogen uptake. This aspect needs to be taken into account if advocating the approach as a generic technique for determining the hydrogen concentration in materials.

As a matter of principle, we agree with the referee that label layers utilized in the resonant neutron reflectometry technique can affect the structural properties – and hence the hydrogen uptake – of a given sample, and that such effects ought to be considered when applying the technique. We have inserted a cautionary sentence in the Discussion section on page 6, as well as the reference by Johansson et al., which indeed illustrates this point very well (See Ref. [50]). We note, however, that the large concentration gradient reported by Johansson et al. was observed close to the surface of a Nb sample without a Pt capping layer, which was added in our case to facilitate penetration and diffusion of H.

In the case at hand, we can indeed rule out any major effects of this kind, for the following reasons:

- In prior neutron reflectometry experiments on magnetic multilayers, including Fe/Nb and Fe/V superlattices (references in text), it was shown that the hydrogen is able to diffuse through the whole structure, going through several Fe layers of thickness 2.6 nm without noticeable concentration gradients. We expect similar behavior for our single 3 nm Co layer.

- As pointed out above, no noticeable differences were found in either the structural properties or the superconducting transition of the upper and lower Nb sublayer. Major differences in hydrogen uptake are therefore unlikely a priori.

- Completely different hydrogen uptake properties of both sublayers would have led to broadening or distortions of the waveguide resonance, which were not observed (inset in Fig. 3d).

- The full reflectometric profile at the end of the hydrogenation experiments, which is now shown as Fig. 2b, does not show any differences in H concentration of both sublayers.

Finally, we point out that the label layer is not needed in principle as explained in the discussion. It is possible to deposit the magnetic (or absorbing) layer directly on the substrate, or omit label layers altogether and use the incoherent scattering of H as the detection channel for the waveguide resonance.

2.c)Hydrogen embrittlement can be profound, even in thin layers. Polycrystalline materials can display large changes upon hydrogen cycling as compared to single crystalline materials, depending on the details of the hydrogen induced volume changes of the material. This needs to be dwelled upon for completeness. It is also worth to highlight the difference in the clamping of the strain field, depending on the crystallographic relation of the substrate and the film.

The reviewer is correct: The hydrogenation process is complex, and embrittlement is a serious concern if hydrogenation is carried out repeatedly. In order to avoid these difficulties in our demonstration experiments, we used only identical pristine samples, which we hydrogenated only once. We added a corresponding explanation to the section "Methods".

2.d) Having magnetic layer in the centre allows for a change in contrast by changing the magnetic direction of that layer. For example, the magnetic direction of the Co layer (with respect to the polarisation axis of the neutrons) can be 0, $\pi/2$, and π . This gives 3 different conditions that can be used to establish better information on the scattering profile and thereby the concentration and the concentration profile within the sample. This can be used to strengthen the communication.

We thank the referee for this very interesting idea. Clearly, the magnetic contrast has the potential to yield further information on the hydrogen density profile in RNR experiments. Exploring this potential is, however, outside the scope of this study and will have to be left for future work. Following guidance from all three referees, we have instead chosen to highlight the complementarity of RNR with conventional neutron reflectometry, which also yields information on the thickness-induced expansion of the active layer. The largest strength of RNR lies in measurements of fast kinetics, and for this purpose a single spin-polarization geometry suffices.

The examples provided above are not complete. Furthermore, depending on which direction the authors decide to take (see under (1) above), different weights need to be given to the issues raised.

We again thank the referee for his/her helpful and constructive comments. Following these comments, as well as those from the other referees, we have refocused the manuscript on the potential of RNR for fast in-situ hydrogenation experiments, which opens new perspectives for in-situ studies of electronic materials and devices, and for the quantitative understanding of the influence of hydrogen intercalation on the electronic properties.

Reviewer #3 (Remarks to the Author):

This is a very well thought out experiment and superbly executed. Physics of the idea is well established and as the authors correctly point out, neutron standing waves provide substantial field enhancement in well fabricated quantum resonators.

This allows one to study thin films by neutron reflectivity near the critical angle and study response from buried layers by exploiting the emission of gamma ray, incoherent scattering, in-plane diffraction and in this article, by looking at the spin flip channel of magnetic scattering in thin films.

The samples made by MBE are very well characterized by various methods. And the authors convincingly demonstrate that Q shift in the resonance of spin flip channel shows very good sensitivity to low levels of hydrogen loading in buried Niobium films.

We thank the reviewer for the thorough and positive assessment of our work.

I have only positive comments to make about this article except for the following. As I mentioned above, these MBE samples are nearly ideally made. Roughness between the films can severely degrade the sharpness of the peaks observed in reflectivity measurements.

At least for the detection of low end of the hydrogen loading quite small subtle shift in Q of resonance peak is required. So it is not clear to me how generally applicable is this broad statement in the abstract "Resonance-enhanced neutron reflectometry thus allows direct, non-destructive measurements of the hydrogen concentration in thin-film structures, with sensitivity high enough for real-time in-situ studies" The paper it self is quite well written.

We have slightly revised the wording in the abstract to emphasize the major strength of the resonance-enhanced method, namely in-situ experiments on systems with fast hydrogen absorption. As demonstrated in the text, the new method indeed increases the hydrogen sensitivity at low concentrations by at least an order of magnitude compared to standard neutron reflectometry.

The only small thing I would point out is that in Fig.3 CAPTION it would be helpful to actually point out which parts (a,b,c,d) belong to which of the two samples that were studied (even though the authors do state that in the text of the article).

The caption of Fig. 3 was revised.

Reviewer #3 - Additional remarks to the Author

Since submitting my review for this paper, I have given it some more thought, especially, after reading other reviewer comments. I have done some simple calculation and, in my view, it is possible to detect hydrogen loading at 1% level that the authors talked about, without doing polarized beam experiments. Unpolarized neutron beam experiments give the same results with better statistics and one does not have to go through the cumbersome routine of making any sophisticated quantum wells.

Here is the simple model for NR calculation.

- 1) Incident medium (air). sld 0
- 2) Pt film 20-50 Ang. sld 6.357
- 3) Nb film 600 Ang. sld 3.92
- 4) Pt film 100 Ang. sld 6.357
- 5) SiO₂ 15Ang. sld 3.4
- 6) Substrate Si sld 2.07
- 7) Assume roughness 5Ang. between all interfaces.

NR calculation for this model gives the first minimum at $\sim .016$ inv. ang. There is steep decline in reflected intensity before reaching this minimum. If one sits at a fixed Q of say $.0155$ inv Ang. and measures the intensity (exactly what the authors have done in their experiment and measure the intensity as a function of sld of Nb, the changes in intensity at this Q are the same percentage as the authors claim in their experiment. In other words, a 1% level of hydrogen is easily detectable.

Other reviewers have commented on the issue of whether the hydrogen detection at 1% level in these special samples is of general interest. My point is that it is not even necessary to do polarized beam experiments (beautiful as they may be) to get the same results. I believe the authors should definitely address this point.

Following the referee's remarks, we have addressed this point in a comprehensive manner by quantitatively comparing fixed-Q scans in NR and RNR experiments. The results are shown in the new Figure 4. It turns out that the Q-position in the NR profiles identified by the referee (near the critical edge or at the first minimum of the Kiessig oscillation) is very narrow in Q, so that the intensity at this point is not simply related to the H concentration. A second position at higher Q turns out to be both broader and more sensitive to the H content. Nonetheless, our calculations show that the sensitivity of the fixed-Q method in RNR experiments to H incorporation still exceeds the optimized NR experiment by an order of magnitude. In addition, the position of the resonance determined in the Q- scan mode of RNR experiments offers a direct, quantitative gauge of the H content. Such scans are much faster than full NR profiles and can be carried out periodically in a real-time experiment to readjust the Q position. At the end of such an experiment, however, a full NR profile is desirable to assess H-induced thickness changes and concentration gradients. In this way, all three neutron methods are complementary.

The reviewer's point is similar to question 14 and 18 of the first referee, and additional information is available in our replies to these questions above. We thank both reviewers for raising this issue, which has led to a substantially improved Discussion section in our manuscript.

REVIEWER COMMENTS

Reviewer #1 (Remarks to the Author):

The authors of the manuscript did a great job and have significantly improved the manuscript according to the comments of the referees. The manuscript is in a much better shape and I recommend publishing in Nature Communications after the following comments have been addressed:

The authors still claim that NR has only been used to quantify hydrogen with amounts of >10 at% and challenged the referee to prove them wrong. As an example, hydrogen concentrations of a few at.% have been determined with specular NR in:

Bannenberg, L. J., Nugroho, F. A. A., Schreuders, H., Norder, B., Trinh, T. T., Steinke, N. J., ... & Dam, B. (2019). Direct comparison of PdAu alloy thin films and nanoparticles upon hydrogen exposure. *ACS applied materials & interfaces*, 11(17), 15489-15497.

The authors show that the Co layer results in homogeneous hydrogenation. Yet, they did not respond to the question about how it would affect the kinetics. As the key advantage of RNR over NR is that it allows for quicker measurements, this advantage is compensated if the contrast layers are affecting the kinetics (as is likely the case). I think this should be addressed in the manuscript as a downside.

Table I is a good starting point for a comparison between the different techniques to quantify hydrogen in thin films. However, it is incomplete. One could e.g. add the need for polarized neutrons, radiation damage, whether the layer expansion is probed simultaneously, or whether it works with complex samples as e.g. multiple layers that hydrogenate.

In the discussion, the authors write that a standard NR measurements takes hours. This is erroneous, even at medium-flux sources/reflectometers and definitely for simple cases as a single Nb film. In this case, a decent measurement can be done in a couple of minutes.

In the simulation of NR data in Figure 4a, please show a typical Q-range measured (i.e. multiple fringes), and an inset displaying the changes at the detailed level the authors presently do. As

opposed to RNR, NR measures more than one fringe and the current way of displaying may be misleading as the discriminative power between subsequent states is higher than illustrated.

The authors talk about the hydrogen 'incorporation' process and hydrogen incorporation. Commonly used terms are hydrogenation process and hydrogenation.

The authors write that the sample remains hydrogenated after the hydrogen pressure was removed ('hydrogenation is irreversible', as e.g. also partly the case for Hf thin films). On which timescale is this observation based? A possible explanation could be that at room temperature, Pt is a very slow catalyst, especially upon desorption. As such, hours/days of oxygen exposure are required for full dehydrogenation. Although the bulk phase diagram suggests that a beta phase is formed, measurements on e.g. Ta (similar phase diagram and also a group V element) show substantial deviations from bulk behavior including a solid solution at room temperature up to the highest hydrogen concentrations measured ($\sim TaH_{0.8}$). The higher hydrogen concentration of the layer can also be explained by these nanoconfinement effects.

Reviewer #2 (Remarks to the Author):

The ideas behind the manuscript are valuable and well worth pursuing. However, there are still shortcomings that hinder appreciation of the ideas and the results presented.

Here below I provide comments on possible shortcomings, inconsistencies and mistakes which I hope the authors will use to increase the impact of their scientific work.

The authors briefly describe a process used to "magnetize(d)" the samples. However, it is not clear how the direction of the magnetisation of the Co layer is during the neutron scattering experiments. This needs to be described and discussed. Furthermore, the inferred moment used in the fitting of the reflectivity data can be compared to the results from the SQUID measurements, demonstrating self consistency in the measurements. Uncertainties can be used to enrich the discussion.

Page 2. Growth of Nb films at temperatures close to ambient results in polycrystalline layers. Growth at elevated temperatures is required to obtain layer by layer growth, and thereby taking an advantage of the matching to the substrate. The statement "The sapphire substrate was chosen because it supports the growth of high-quality Nb films ..." can therefore be viewed as misleading.

Page 6. Dangling modifier: “Neutron-based methods, in contrast, are unaffected by radiation damage and allow in-situ loading experiments in H...” it is the sample which can be affected, not the method.

In figure 4.b, replace “prisitne” by “pristine”.

Page 6. Could it be that you made a mistake when writing “The diffraction patterns indicate a polycrystalline texture of the niobium film, with preferential orientation along the (110) plane.”?

Page 6. You write: “Hysteresis loops obtained by SQUID magnetometry at room temperature showed a saturation magnetic moment approaching the bulk value of 1.5 kG.” Please provide the numbers, furthermore, to say “... approaching the bulk ...” is not appropriate.

Page 6. You write: “Transport measurements using a four-probe device described in previous work [42] revealed a superconducting transition at \sim 7 K, slightly lower than in the bulk due to the proximity of the Co layer [56].”. The causal relation is not provided and the statement is therefore speculative. Strain state, finite size effects, impurities etc can all contribute to the reduction in in the superconducting transition.

Page 6. You write: “The resulting thicknesses were ... 270 Å for each Nb layer”. This is not consistent with the numbers provided in table S1. Missing uncertainties and error analysis.

Page 7. You write: “ The corresponding roughness parameters of around 4 Å for Nb, 10 Å for Co and 12 Å for Pt are well below the layer thicknesses. ”. Co is grown on Nb, and one of the Nb layers is grown on Co. How is it possible that the Co and the Nb have different “roughness” at the same interface?

In table I you specify the sensitivity of NRA to be 0.1 at.%. The N-15 technique has substantially lower detection limit. Adding “<” (i.e. below 0.1 ...) is an easy way to correct this.

In the supplementary information you specify the Nb layers to be 268 and 276 Å prior to hydrogen loading (not the same thickness, see comment above). After the hydrogen loading the layer thicknesses are specified to the 299 and 290 Å. Hence the first layer can be inferred to be expanded

by 31 Å by exposure to hydrogen, while the second layer can be inferred to be expanded by 14 Å. Taking these numbers literally, one can conclude that there is a factor of 2 difference in the expansion. This is somewhat problematic. If the layers have the same initial texture and strain state, this would imply that the concentration could be expected to differ by a factor of 2, which is in clear contradiction to the statements on the concentrations in the layers.

Reviewer #3 (Remarks to the Author):

Since submitting my last report on this study, I have given it some more thought.

I still believe that it is possible to carry out measurement of few atomic % H loading in Nb films using regular Neutron Reflectivity and as fast or faster than their spin polarized RNR measurements.

I want to go back to the model that I suggested in my last report.

Here is a slightly modified version of that model, although the conclusions made previously do not change.

- 1) Incident medium sld 0
- 2) Pt film 40Ang. Sld 5.7 (I have changed according to their table S1)
- 3) Nb film 550 Ang. Sld 3.92 (conforms with total Nb thickness in able S1)
- 4) Pt film 100 Ang. Sld 5.7
- 5) SiO₂ 100 Ang. Sld 3.4 (This does not need to be there, but I am told that if one wants to sputter Pt on a Si wafer with natural 10-15 Ang. SiO₂, then one can get quite a bit of intermixing between Si and Pt, thereby increasing the interface roughness)
- 6) Substrate Si Sld 2.07
- 7) 7) Assume 5Ang. Roughness between layers.

Calculating NR for this model one gets the first minimum at .016192 inv. Ang.

At 5% H loading (Nb95H5) sld of Nb layer is 3.807 (assuming no change in density and thickness) and the first minimum shifts to .01599. A shift of .0002 inv. Ang. This is in line with the expected shift in their sample according to Eq. (4) in their paper.

Calculating the reflectivity at a Q of .0155 we get

No H loading (Nb sld 3.92). $R = .275$

5% H loading (Nb sld 3.807) $R = .155$

Using their Eq(6) and assuming initial beam of 1500/sec, measuring time for this is .051 sec., same order of magnitude or smaller time than in their RNR measurements shown in their Fig. (4).

I have not included any H incorporation in the Pt films, but including that would have a minimal effect. These counting times would surely be longer in real world samples where one has more roughness between interfaces and instrumental resolution has to be accounted for, but the effect of these variables should not be very large.

In any case, my point is that there is no need to do RNR spin polarized measurements to get at H loading of few atomic % in thin Nb films. One has to carefully create a sample of a suitable sld profile to be able to do this. In fact, the sld profile that I have used does have a quasi-resonant cavity between the top and bottom Pt layers. In Fig (4) the authors have calculated the NR for their model at low Q and correctly show that the counting times for NR are longer than for RNR. However, if they use a lower sld substrate in their samples, the first minimum would be pushed to a lower Q, $\sim .015$ inv. Ang. Where reflectivities are much higher and counting time lower.

Spin polarized RNR part of the experiment is done very well. But, to repeat myself, it is not required.

Sushil Satija

NIST Center for Neutron Research

In the spirit of transparent review process, I do not mind being open about my identity. I respect the authors expertise and the good work that they do and will be happy to hear their reply if they think my analysis is wrong.

REVIEWER COMMENTS

Reviewer #1 (Remarks to the Author):

The authors of the manuscript did a great job and have significantly improved the manuscript according to the comments of the referees. The manuscript is in a much better shape and I recommend publishing in Nature Communications after the following comments have been addressed:

We thank the Reviewer for the positive assessment on the new version of the manuscript.

The authors still claim that NR has only been used to quantify hydrogen with amounts of >10 at% and challenged the referee to prove them wrong. As an example, hydrogen concentrations of a few at.% have been determined with specular NR in: Bannenbergh, L. J., Nugroho, F. A. A., Schreuders, H., Norder, B., Trinh, T. T., Steinke, N. J., ... & Dam, B. (2019). Direct comparison of PdAu alloy thin films and nanoparticles upon hydrogen exposure. ACS applied materials & interfaces, 11(17), 15489-15497.

We thank the Reviewer for the reference, which we added as Ref. 23. According to Fig. 3a in this article, a ~ 5% concentration of hydrogen can be measured with statistical confidence outside the error bar. We included this value in the new version of the manuscript.

The authors show that the Co layer results in homogeneous hydrogenation. Yet, they did not respond to the question about how it would affect the kinetics. As the key advantage of RNR over NR is that it allows for quicker measurements, this advantage is compensated if the contrast layers are affecting the kinetics (as is likely the case). I think this should be addressed in the manuscript as a downside.

A control NR measurement on a 50 nm thick Nb film without a Co layer did not show substantial differences in hydrogenation kinetics (within the error limits afforded by non-resonant NR): Like in the RNR experiment, ~ 85% hydrogenation was reached after ~ 700 minutes. This is consistent with the observation of homogeneous hydrogenation in the RNR experiment, as gross differences in hydrogen diffusion kinetics across the Co layer would have resulted in substantial inhomogeneity. We have added a corresponding sentence to the main text (page 4, right column). In general, we agree with the Reviewer that possible effects of label layers on the diffusion kinetics should be considered, and we have added a corresponding statement to the caveats mentioned on page 5, right column.

However, this does not constitute a general drawback of the RNR, since the label layer can be shifted below the layer of interest or not used at all, as discussed elsewhere in the manuscript and in the prior correspondence with the reviewers. On the other hand, the design we have used here also offers distinct opportunities, as RNR with a high concentration of neutrons on an atomically thin magnetic layer can be used to map out the magnetic ground state of spintronic devices during hydrogenation. We have added a corresponding sentence to the “Neutron waveguide resonance” section.

Table I is a good starting point for a comparison between the different techniques to quantify hydrogen in thin films. However, it is incomplete. One could e.g. add the need for polarized neutrons, radiation damage, whether the layer expansion is probed simultaneously, or whether it works with complex samples as e.g. multiple layers that hydrogenate.

We thank the Reviewer for these suggestions. The radiation damage and expansion issues have been added to the table. As we explain in the text, RNR does not necessarily require polarized neutrons, since gamma or alpha radiation, channelling, transmittance or incoherent scattering can be used to detect the resonance. In the case of multiple layers, RNR can be still used, for example by utilizing the principles of magnetic waveguides (see ref. [40] for details).

In the discussion, the authors write that a standard NR measurements takes hours. This is erroneous, even at medium-flux sources/reflectometers and definitely for simple cases as a single Nb film. In this case, a decent measurement can be done in a couple of minutes.

We agree that this generic statement was inadequate and have corrected the text accordingly.

In the simulation of NR data in Figure 4a, please show a typical Q-range measured (i.e. multiple fringes), and an inset displaying the changes at the detailed level the authors presently do. As opposed to RNR, NR measures more than one fringe and the current way of displaying may be misleading as the discriminative power between subsequent states is higher than illustrated.

Indeed, the NR measures multiple fringes and the sensitivity $R-R_H$ in the first approximation is the same for all of them. However, we note that a typical reflectivity curve falls off as Q^{-4} , so that the measurement time for the effect of hydrogenation according to Eq. 6 increases by orders of magnitude compared to the Q-range near total reflection. In addition, the presentation of data on such a large Q-scale makes it extremely difficult to see the changes in resonance and oscillation when hydrogen is injected. Anyway, for the sake of completeness we have added the requested simulation to the Supplemental Materials (Fig. S4).

The authors talk about the hydrogen ‘incorporation’ process and hydrogen incorporation. Commonly used terms are hydrogenation process and hydrogenation.

We have changed our wording accordingly.

The authors write that the sample remains hydrogenated after the hydrogen pressure was removed (‘hydrogenation is irreversible’, as e.g. also partly the case for Hf thin films). On which timescale is this observation based? A possible explanation could be that at room temperature, Pt is a very slow catalyst, especially upon desorption. As such, hours/days of oxygen exposure are required for full dehydrogenation. Although the bulk phase diagram suggests that a beta phase is formed, measurements on e.g. Ta (similar phase diagram and also a group V element) show substantial deviations from bulk behavior including a solid solution at room temperature up to the highest

hydrogen concentrations measured (\sim TaH0.8). The higher hydrogen concentration of the layer can also be explained by these nanoconfinement effects.

We have observed that the samples are still hydrogenated several months after the first loading, both with PNR and with transport measurements. We have added a corresponding comment to the Results section (page 4, right column).

Reviewer #2 (Remarks to the Author):

The ideas behind the manuscript are valuable and well worth pursuing. However, there are still shortcomings that hinder appreciation of the ideas and the results presented. Here below I provide comments on possible shortcomings, inconsistencies and mistakes which I hope the authors will use to increase the impact of their scientific work.

We thank the Reviewer for the careful reading of our manuscript and for the constructive comments.

The authors briefly describe a process used to “magnetize(d)” the samples. However, it is not clear how the direction of the magnetisation of the Co layer is during the neutron scattering experiments. This needs to be described and discussed.

We have included further details in the Methods about the layer magnetization direction, explaining that at the remanent field of 5 Oe used during our experiments the magnetization is aligned along the easy axis direction, e.g. around the diagonal of the sample, hence ensuring non-collinearity to the applied field and a subsequent spin-flip signal, following the work of Wolff et. al [36]. We also observed from PNR fitting that the magnetization direction is robust upon H-absorption and it is still along the easy axis even after 85% of H atoms are absorbed. We added the angle of magnetization in the summary of parameters extracted from fitting in Table S3.

Furthermore, the inferred moment used in the fitting of the reflectivity data can be compared to the results from the SQUID measurements, demonstrating self consistency in the measurements. Uncertainties can be used to enrich the discussion.

Both PNR and SQUID measurements are quantitatively consistent. The corresponding values and uncertainties are now stated in the Methods section (page 7, left column). We have also modified Table S3 to include the magnetization inferred from PNR in the same units of kG as the one obtained from SQUID (Fig. S1d), showing the consistency of the two different measurements. The uncertainties of the parameters were added.

Page 2. Growth of Nb films at temperatures close to ambient results in polycrystalline layers. Growth at elevated temperatures is required to obtain layer by layer growth, and thereby taking an advantage of the matching to the substrate. The statement “The sapphire substrate was chosen because it supports the growth of high-quality Nb films ...” can therefore be viewed as misleading.

We agree with the Reviewer and have cut the sentence.

Page 6. Dangling modifier: “Neutron-based methods, in contrast, are unaffected by radiation damage and allow in-situ loading experiments in H...” it is the sample which can be affected, not the method.

In figure 4.b, replace “prisitne” by “pristine”.

We thank the Reviewer and have modified the text and the figure legend.

Page 6. Could it be that you made a mistake when writing “The diffraction patterns indicate a polycrystalline texture of the niobium film, with preferential orientation along the (110) plane.”?

We are at a loss to find an error in this sentence. Polycrystalline growth with preferential (110) direction is typical for niobium films at room temperature, which was shown in many studies, including our prior work [56].

Page 6. You write: “Hysteresis loops obtained by SQUID magnetometry at room temperature showed a saturation magnetic moment approaching the bulk value of 1.5 kG.” Please provide the numbers, furthermore, to say “... approaching the bulk ...” is not appropriate.

The moment of 1 kG was already shown in the Supplementary Materials, Figure S1d. In addition, we have modified the main text in the Methods section with precise values, following the Reviewer’s suggestion.

Page 6. You write: “Transport measurements using a four-probe device described in previous work [42] revealed a superconducting transition at \sim 7 K, slightly lower than in the bulk due to the proximity of the Co layer [56].”. The causal relation is not provided and the statement is therefore speculative. Strain state, finite size effects, impurities etc can all contribute to the reduction in in the superconducting transition.

*We agree with the Reviewer that the suppression of T_c can in general be caused by several reasons, including strain, finite size effects etc. We note, however, that in our prior work [56], a 50nm niobium layer deposited under similar conditions showed a T_c of 9.1K, much closer to the bulk value of 9.4K. This allows us to rule out impurities and conclude with some degree of certainty that the drop in T_c is most likely caused by the proximity effect. However, the referee is right in stating that we have not demonstrated this causal relationship, and we do not wish to go deeply into this issue because the presence of an abrupt superconducting transition in the niobium layer was used by us solely for the purpose of demonstrating the homogeneity of the film. We therefore changed the text to “Transport measurements (...) revealed a sharp superconducting transition at \sim 7 K, which is slightly lower than in the bulk, **possibly** due to the proximity of the Co layer.”*

Page 6. You write: “The resulting thicknesses were ... 270 Å for each Nb layer” . This is not consistent with the numbers provided in table S1. Missing uncertainties and error analysis.

We have modified the text by adding all relevant uncertainties.

Page 7. You write: “The corresponding roughness parameters of around 4 Å for Nb, 10 Å for Co and 12 Å for Pt are well below the layer thicknesses.”. Co is grown on Nb, and one of the Nb layers is grown on Co. How is it possible that the Co and the Nb have different “roughness” at the same interface?

In the GenX software that we used, the roughness is calculated at the upper interface of a layer. Starting from the substrate, we obtain a roughness of 2 Å at the interface between Al₂O₃/Nb, then a roughness

of 4 Å between Nb1/Co, then 10 Å roughness between Co/Nb2 and so on. We have added the corresponding explanation to the description of table S3.

In table I you specify the sensitivity of NRA to be 0.1 at.%. The N-15 technique has substantially lower detection limit. Adding “<” (i.e. below 0.1 ...) is an easy way to correct this.

We thank the Reviewer for the comment and have added the “<”.

In the supplementary information you specify the Nb layers to be 268 and 276 Å prior to hydrogen loading (not the same thickness, see comment above). After the hydrogen loading the layer thicknesses are specified to the 299 and 290 Å. Hence the first layer can be inferred to be expanded by 31 Å by exposure to hydrogen, while the second layer can be inferred to be expanded by 14 Å. Taking these numbers literally, one can conclude that there is a factor of 2 difference in the expansion. This is somewhat problematic. If the layers have the same initial texture and strain state, this would imply that the concentration could be expected to differ by a factor of 2, which is in clear contradiction to the statements on the concentrations in the layers.

We have added the uncertainties to all values obtained by fitting. As can be seen from the updated table S3, the thicknesses of the upper and bottom Nb layers are equal within the error bar.

Reviewer #3 (Remarks to the Author):

Since submitting my last report on this study, I have given it some more thought. I still believe that it is possible to carry out measurement of few atomic % H loading in Nb films using regular Neutron Reflectivity and as fast or faster than their spin polarized RNR measurements.

I want to go back to the model that I suggested in my last report.

Here is a slightly modified version of that model, although the conclusions made previously do not change.

- 1) Incident medium sld 0
- 2) Pt film 40Ång. Sld 5.7 (I have changed according to their table S1)
- 3) Nb film 550 Ång. Sld 3.92 (conforms with total Nb thickness in able S1)
- 4) Pt film 100 Ång. Sld 5.7
- 5) SiO₂ 100 Ång. Sld 3.4 (This does not need to be there, but I am told that if one wants to sputter Pt on a Si wafer with natural 10-15 Ång. SiO₂, then one can get quite a bit of intermixing between Si and Pt, thereby increasing the interface roughness)
- 6) Substrate Si Sld 2.07
- 7) Assume 5Ång. Roughness between layers.

Calculating NR for this model one gets the first minimum at .016192 inv. Ång.
At 5% H loading (Nb₉₅H₅) sld of Nb layer is 3.807 (assuming no change in density and thickness) and the first minimum shifts to .01599. A shift of .0002inv. Ång. This is in line with the expected shift in their sample according to Eq. (4) in their paper.

Calculating the reflectivity at a Q of .0155 we get

No H loading (Nb sld 3.92). R= .275

5% H loading (Nb sld 3.807) $R = .155$

Using their Eq(6) and assuming initial beam of 1500/sec, measuring time for this is .051 sec., same order of magnitude or smaller time than in their RNR measurements shown in their Fig. (4).

I have not included any H incorporation in the Pt films, but including that would have a minimal effect. These counting times would surely be longer in real world samples where one has more roughness between interfaces and instrumental resolution has to be accounted for, but the effect of these variables should not be very large.

In any case, my point is that there is no need to do RNR spin polarized measurements to get at H loading of few atomic % in thin Nb films. One has to carefully create a sample of a suitable sld profile to be able to do this. In fact, the sld profile that I have used does have a quasi-resonant cavity between the top and bottom Pt layers. In Fig (4) the authors have calculated the NR for their model at low Q and correctly show that the counting times for NR are longer than for RNR. However, if they use a lower sld substrate in their samples, the first minimum would be pushed to a lower Q, $\sim .015$ inv. Ang. Where reflectivities are much higher and counting time lower.

Spin polarized RNR part of the experiment is done very well. But, to repeat myself, it is not required.

Sushil Satija
NIST Center for Neutron Research

In the spirit of transparent review process, I do not mind being open about my identity. I respect the authors expertise and the good work that they do and will be happy to hear their reply if they think my analysis is wrong.

We would like to thank the Reviewer for such a detailed study of our results. We are pleased to note that the structure which the reviewer came up with as a result of his optimization is a resonator. Moreover, we have calculated the neutron density in this structure (see panel d of the Figure below) and find that the mentioned $Q \sim 0.016 \text{ \AA}^{-1}$ with minimum measurement time coincides with the maximum of the resonant field. The resonator with low-SLD substrate suggested by the Reviewer is similar to the so-called neutron interference filters used to monochromatize ultracold neutrons (see the new Ref. 38). For these "UCN-filters", transmission can be used to detect the resonance. In this case, the RNR signal at Q_{res} is tending to one, so that even faster measurement of hydrogen loading (compared to tracing the dip in the NR curve) is possible. Note, however, that this particular design has its drawbacks, including the need to carefully separate transmitted and direct beams, the difficulty of working with negative-SLD films etc. In any case, we again want to thank the Reviewer for his thoroughness, since it allows us to point out one more detection channel of the resonance, namely transmission. We have augmented the section "Neutron waveguide resonances" correspondingly.

Finally, the issue of the magnetic layer was already raised by the first Reviewer in the previous round of correspondence, and we refer to our reply to this query. Briefly, we reiterate that a magnetic layer is not necessarily needed, and other detection channels not requiring polarized neutrons can be used. The exchange with the Reviewer has also suggested an interesting secondary application of our design, which will allow one to study the effect of hydrogen on magnetism in atomically thin magnetic layers. A corresponding sentence has been added to the theoretical section.

Reviewer #1 (Remarks to the Author):

The authors have answered the points raised in a satisfactory manner, and adjusted the manuscript correspondingly (with the exception of the point below.). I therefore recommend publishing in Nature Communications.

Comment:

The authors write: 'Indeed, the NR measures multiple fringes and the sensitivity R-RH in the first approximation is the same for all of them. However, we note that a typical reflectivity curve falls off as Q^{-4} , so that the measurement time for the effect of hydrogenation according to Eq. 6 increases by orders of magnitude compared to the Q-range near total reflection. In addition, the presentation of data on such a large Q-scale makes it extremely difficult to see the changes in resonance and oscillation when hydrogen is injected.'

The authors are correct that the reflectivity typically falls off with Q^4 . However, especially for TOF neutron reflectometry, this is compensated by the orders of magnitude higher intensity at the shorter wavelengths. As such, these differences can be easily observed and are used to probe the changes in SLD.

Reviewer #2 (Remarks to the Author):

The authors have used the feedback from the reviewers in a constructive way and thereby improved the manuscript to the level of acceptance.

Reviewer #3 (Remarks to the Author):

Unfortunately, I disagree with the authors.

For example, Fig. S4 is in my opinion is (even though quite unintentionally, I am sure) a bit misleading. The authors calculate the regular NR and counting times for detecting hydrogen in their film. As far as it goes, that is accurate. However, as I mentioned in my previous review, if you change the substrate to Si (With SLD of 2.07), the picture changes considerably. First minimum in reflectivity shifts from $\sim .022$ to $\sim .017$ and it becomes quite easy to detect a 5% loading of Hydrogen in much less than a second. In fact, a single film of Nb on Si of 55nm on Si gives the first minimum at $.0175$. And I maintain that it is easily possible to detect a 5% H loading in this single film in less than half a second and 1% level within a minute.

My contention is that RNR is not required for this level of Hydrogen detection.

In their response to my last review the authors write that

"Note, however, that this particular design has

its drawbacks, including the need to carefully separate transmitted and direct beams, the difficulty of

working with negative-SLD films etc."

In this regard I would like to posit that people in NR have been very easily separating the transmitted and direct beam from the very beginning with very tightly collimated beams. Without the tightly collimated beams it would not be possible to measure reflectivities at all! Also, I do not quite understand the comment about negative_SLD films.

My final comments.

1) The authors correctly demonstrate that there is a spin flip channel in an appropriately constructed resonance structure which can be used to detect low levels of H loading in thin films. However, as I have tried to show here, a resonant structure is not needed for such a measurement.

2) They have also already shown the feasibility of exploiting the RNR spin flip channel for magnetic film studies (Their ref. 41, which is missing a page No.BTW). Plus, in the light of the fact that RNR is not needed for this level of H detection, I am not quite sure this paper fits in for publication in Nature Communications. Although I would defer to the Editors for final decision.

Sushil Satija

REVIEWER COMMENTS

Reviewer #1 (Remarks to the Author):

The authors have answered the points raised in a satisfactory manner, and adjusted the manuscript correspondingly (with the exception of the point below.). I therefore recommend publishing in Nature Communications.

We thank the reviewer for the positive assessment on the new version of the manuscript.

The authors write: 'Indeed, the NR measures multiple fringes and the sensitivity $R-RH$ in the first approximation is the same for all of them. However, we note that a typical reflectivity curve falls off as Q^{-4} , so that the measurement time for the effect of hydrogenation according to Eq. 6 increases by orders of magnitude compared to the Q -range near total reflection. In addition, the presentation of data on such a large Q -scale makes it extremely difficult to see the changes in resonance and oscillation when hydrogen is injected.'

The authors are correct that the reflectivity typically falls off with Q^4 . However, especially for TOF neutron reflectometry, this is compensated by the orders of magnitude higher intensity at the shorter wavelengths. As such, these differences can be easily observed and are used to probe the changes in SLD.

We agree with the referee that TOF reflectometers have certain advantages due to their coverage of an extended range of Q . However, direct calculations (see Fig. 1 below) show that the quantity $(R-RH)^2$ in the denominator of Eq. (6), which parameterizes the sensitivity to hydrogen, decreases by orders of magnitude with increasing Q . Measurements of higher-order Kiessig fringes at a TOF spectrometer therefore do not have any significant influence on the hydrogen sensitivity.

Fig. 1. Sensitivity parameter $(R-RH)^2$ for the Pt(3nm)/Nb(55nm)//Al₂O₃ structure upon 5% hydrogenation.

Reviewer #3 (Remarks to the Author):

Unfortunately, I disagree with the authors. For example, Fig. S4 is in my opinion is (even though quite unintentionally, I am sure) a bit misleading. The authors calculate the regular NR and counting times for detecting hydrogen in their film. As far as it goes, that is accurate. However, as I mentioned in my previous review, if you change the substrate to Si (With SLD of 2.07), the picture changes considerably. First minimum in reflectivity shifts from ~ 0.022 to ~ 0.017 and it becomes quite easy to detect a 5% loading of Hydrogen in much less than a second. In fact, a single film of Nb on Si of 55nm on Si gives the first minimum at 0.0175 . And I maintain that it is easily possible to detect a 5% H loading in this single film in less than half a second and 1% level within a minute. My contention is that RNR is not required for this level of Hydrogen detection.

Following these comments, we have prepared a new version of Fig. 4 in the main text¹ that fully addresses the reviewer's concerns. Panels c and d of this figure display the results of simulations for Nb films on silicon substrates, as suggested by the reviewer. (Note these calculations take into account all relevant experimental conditions at the NREX instrument, where most of our measurements were taken, including the instrumental resolution and the losses incurred due to the need for spin-polarized beams in RNR experiments.) They confirm that NR measurements can detect 5% hydrogen incorporation in less than a minute by changing the substrate to silicon, as contended by the reviewer (panel c), although the calculated sensitivity enhancement is not as large as the one that can be realized in RNR for the film on sapphire (panel b). Crucially, an additional large enhancement of the hydrogen sensitivity can be realized by creating a waveguide structure via insertion of a thin Pt layer between the silicon substrate and the niobium overlayer (panel d). Note that Pt deposition is required anyway for the capping layer, so that insertion of the layer will not create additional complications, and that spin-flip scattering is not required. With the resonant enhancement, the calculated detection time for 5% hydrogen is only 40 msec. RNR thus enables an additional order-of-magnitude sensitivity enhancement for hydrogen, on top of the one realized by optimizing the NR setup. In addition to this example and the one presented in Fig. 4b (where the detection time is 60 msec), we have added two examples to the Supplementary Materials (Fig. S5) which demonstrate detection times in different waveguide structures with a different detection scheme (a Gd absorber) and SF scattering from an Fe layer inserted at the substrate interface. The versatility of RNR illustrated by these examples provides confidence that this technique will be widely adopted in the community.

In the revised version of the manuscript, we have modified the messaging in several instances (marked in red) to avoid any impression that specific numerical sensitivity levels are out of reach for conventional NR. Rather, we stress the potential for additional large enhancements by creating waveguide structures, and the multiple options that are available for the realization of such structures.

In their response to my last review the authors write that "Note, however, that this particular design has its drawbacks, including the need to carefully separate transmitted and direct beams, the difficulty of working with negative-SLD films etc." In this regard I would like to posit that people in NR have been very easily separating the transmitted and direct beam from the very beginning with very tightly collimated beams. Without the tightly collimated beams it would not be possible to measure reflectivities at all! Also, I do not quite understand the comment about negative_SLD films.

¹ As Supplementary Fig. S4 was inserted open specific request of reviewer 1, we are reluctant to replace it. We are confident, however, that the potentially (and of course, unintentionally) misleading impression the reviewer will be pre-empted by our new Fig. 4 and by the revised wording of our manuscript.

These remarks are now superseded by simulations in Fig. 4 and the corresponding discussion. We had meant to say that tight collimation unavoidably reduces the incoming intensity, and that the associated intensity tradeoff needs to be carefully considered. Moreover, substrates with low SLD suitable for transmission measurements are not available for every thin-film system. (The remark about negative-*Q* substrates was a digression, which was unhelpful as we now realize). Clearly, however, the experimental geometry suggested by the reviewer does offer an intensity advantage in some cases. This is now fully acknowledged in a general sense in the introduction of the manuscript (page 1, right column), in the discussion (page 5, right column), and finally in the context of the concrete case suggested by the reviewer in Fig. 4 and associated discussion on page 6.

My final comments.

1) The authors correctly demonstrate that there is a spin flip channel in an appropriately constructed resonance structure which can be used to detect low levels of H loading in thin films. However, as I have tried to show here, a resonant structure is not needed for such a measurement.

We have experimentally demonstrated the wave-guide enhanced sensitivity enhancement using spin-flip scattering; such a demonstration is crucial in order to establish the credibility of this approach. At the same time, we have stated from the beginning that spin-flip scattering is only one of several options to realize waveguide enhancement, and we have now backed up this assertion by detailed numerical calculations for transmission measurements (the geometry suggested by the reviewer, Fig. 4) and for a neutron absorber (Fig. S5).

We agree with the reviewer that it might be possible to reach any specific numerical sensitivity target by optimizing the design of conventional reflectometry experiments, without resorting to waveguide structures. In this sense, some of the wording in the previous version of our manuscript may have given a misleading impression. In the new version, we have pre-empted any such impression by including a new figure that explicitly addresses the issue raised by the reviewer, and by emphasizing the opportunity for additional substantial enhancements by introducing waveguide effects even in structures and geometries that have already been optimized otherwise.

2) They have also already shown the feasibility of exploiting the RNR spin flip channel for magnetic film studies (Their ref. 41, which is missing a page No.BTW). Plus, in the light of the fact that RNR is not needed for this level of H detection, I am not quite sure this paper fits in for publication in Nature Communications. Although I would defer to the Editors for final decision.

Waveguide resonances are well known and have been used for many decades in neutron scattering for various different purposes, as discussed at length in the introduction of our paper. Ref. 41, which introduced “switchable” waveguides in magnetic multilayers and devices, has no essential relationship to our current manuscript. In particular, the structures investigated in Ref. 41 are based on a completely different principle, where the magnetic layers actually form the waveguide structure (rather than serving as a detector of the neutron density, as in our present work). The novelty of the current work is to exploit resonators for hydrogen detection and quantification with enhanced sensitivity. In particular, we have presented a simple new formula (Eq. 4) that relates the resonance position to the absolute H content. The RNR scheme we introduced will enable novel real-time in-situ studies of the electronic properties of thin-film structures under hydrogen exposure that could not be anticipated based on any prior literature.

REVIEWERS' COMMENTS

Reviewer #1 (Remarks to the Author):

Dear Dr Margherita Citroni,

Thank you for your inquiry. I have read the revised version of the manuscript entitled "Resonant neutron reflectometry for hydrogen detection".

To start with, the development of new/improved measurement techniques is crucial for the advancement of science. I therefore greatly appreciate the work performed by the authors and this was one of the reasons why I recommended publishing the manuscript before.

In your request, you specifically ask whether I think 'the authors are able to demonstrate, at least in principle, an improvement in sensitivity with the proposed technique compared to the established neutron reflectivity techniques'. My opinion is mixed on this issue. Although the authors are more realistic about the capabilities of conventional neutron reflectometry (NR) in this version of the manuscript than in previous ones, they, as Referee 3 also points out, underquote its performance: e.g. the lowest concentration limit that conventional NR can detect in Table 1 is overly pessimistic. As noted before, +/- 1 at.% H has been detected as I noted before, see, e.g., ref. 23. I do wish to note this highly depends on the sample and measurement times, but this also applies for the RNR method presented here (in the introduction the authors do acknowledge that <5% is achievable following previous comments).

On the other hand, I do think that RNR actually has the potential to perform measurements of limited changes in H concentration in a faster way than conventional NR, and, as a result, for kinetic experiments, may achieve a higher sensitivity. Note however that this will only be possible if the sample is optimized to the even tighter requirements for RNR.

The second point you ask my opinion on is whether the 'method's design itself is particularly valuable for further development of sensing techniques'. I do believe that development of methods is extremely valuable. Although it is a priori hard to predict whether a method will be applied extensively or not, this manuscript documents a technique that has certain advantages, particularly shorter measurement times that can be relevant for kinetic studies (especially at reactor neutron sources), over techniques that are currently applied in the literature. As the authors point out in

their manuscript, there are other ways to use a resonant approach to detect hydrogen which will can be explored in further studies.

I hope these remarks help you in making the final decision.

p.s. I disagree with the author's response to my remark in their rebuttal. They write 'We agree with the referee that TOF reflectometers have certain advantages due to their coverage of an extended range of Q. However, direct calculations (see Fig. 1 below) show that the quantity $(R-RH)^2$ in the denominator of Eq. (6), which parameterizes the sensitivity to hydrogen, decreases by orders of magnitude with increasing Q. Measurements of higher-order Kiessig fringes at a TOF spectrometer therefore do not have any significant influence on the hydrogen sensitivity.' In this analysis, the authors have probably overlooked the fact that the intensity at the higher-order Kiessig fringes is also orders of magnitude higher, as this corresponds to the shorter wavelengths used rather than the less-intense longer wavelength used to probe changes in reflectivity at smaller Q-values. As such, this compensates the rapid decrease of the quantity $(R-RH)^2$ in the denominator. Consider e.g. the hydrogenation of a Pd-capped Ta film in Fig 6b in <https://doi.org/10.1016/j.snb.2018.12.029>: the sensitivity to hydrogen mostly comes from the first and second fringe and not from changes near the critical edge.

Reviewer #3 (Remarks to the Author):

Unfortubately, I still disagree with the authors that RNR is required to detect hydrogen loading of very small amounts in thin films.

I have chosen to do NR calculation with the following two models.

- 1) Pt(3)/Nb(55)/Si and
- 2) Pt(3)/Nb(55)/Pt(10)/Si (their wave guide structure model)

Both with and without 5% H loading.

I assume the following parameters.

Sld Nb = 3.92, Si = 2.068, Pt (6.36), Sld Nb95H5 = 3.643 [Here I assume that density of Nb is 95% of pure Nb]

All of the interface roughnesses in both models are kept at 5Ang.

Here are the results that I get for Tmin using their Eq. 6 and assuming beam intensity to be 1500/sec.

For the Pt(3)/Nb(55)/Si model I get Tmin= .0392sec @ Q=.0155.

For the Pt(3)/Nb(55)/Pt(10)/Si model I get Tmin= .0144sec @Q=.0155

I am not quite sure how the authors get a Tmin of .25sec shown in Fig. 4c. I would urge them to check their calculation again. I have checked it several times and I am quite sure Tmin of .0392sec is correct.

In any case the difference in the time between to cases is a factor of 3 and not 6 as claimed by the authors.

In the second case the total thickness is 10nm larger so the whole NR curve is shifted towards lower Q and that is one reason that Tmin shifts to lower values.

I have checked this by increasing the thickness of the Nb film in the first model to 65nm (Pt(3)/Nb(65)/Si) and now Tmin= .0223sec at Q=.0153. This is only 50% higher than model 2.

My sense after looking at this issue for some time is that whether one has a wave guide structure for RNR or the first model with only Nb film on Si, in both cases the measurement is being done at a low enough Q where the reflectivity is quite high. When reflectivity is already this high it is difficult to get gains of order of magnitude. The trick is to make a structure where the first minimum occurs at low enough Q and that can be easily done with NR.

If the authors disagree with my calculations, I will be happy to provide them with detailed sld profiles and calculated reflectivities.

Response to reviewers:

Reviewer #1

To start with, the development of new/improved measurement techniques is crucial for the advancement of science. I therefore greatly appreciate the work performed by the authors and this was one of the reasons why I recommended publishing the manuscript before.

We thank the reviewer for her/his thorough and constructive review, and for the positive recommendation.

In your request, you specifically ask whether I think ‘the authors are able to demonstrate, at least in principle, an improvement in sensitivity with the proposed technique compared to the established neutron reflectivity techniques’. My opinion is mixed on this issue. Although the authors are more realistic about the capabilities of conventional neutron reflectometry (NR) in this version of the manuscript than in previous ones, they, as Referee 3 also points out, underquote its performance: e.g. the lowest concentration limit that conventional NR can detect in Table 1 is overly pessimistic. As noted before, +/- 1 at.% H has been detected as I noted before, see, e.g., ref. 23. I do wish to note this highly depends on the sample and measurement times, but this also applies for the RNR method presented here (in the introduction the authors do acknowledge that <5% is achievable following previous comments).

We have revised the introduction and conclusion once again to make sure that the potential of conventional neutron reflectometry is not understated. We have also added a footnote to Table I (with a citation of Ref. [23]) to acknowledge that 1% hydrogen sensitivity has indeed been obtained by conventional neutron reflectometry.

p.s. I disagree with the author’s response to my remark in their rebuttal. They write ‘We agree with the referee that TOF reflectometers have certain advantages due to their coverage of an extended range of Q. However, direct calculations (see Fig. 1 below) show that the quantity $(R-R_H)^2$ in the denominator of Eq. (6), which parameterizes the sensitivity to hydrogen, decreases by orders of magnitude with increasing Q. Measurements of higher-order Kiessig fringes at a TOF spectrometer therefore do not have any significant influence on the hydrogen sensitivity.’ In this analysis, the authors have probably overlooked the fact that the intensity at the higher-order Kiessig fringes is also orders of magnitude higher, as this corresponds to the shorter wavelengths used rather than the less-intense longer wavelength used to probe changes in reflectivity at smaller Q-values. As such, this compensates the rapid decrease of the quantity $(R-R_H)^2$ in the denominator. Consider e.g. the hydrogenation of a Pd-capped Ta film in Fig 6b in <https://doi.org/10.1016/j.snb.2018.12.029>: the sensitivity to hydrogen mostly comes from the first and second fringe and not from changes near the critical edge.

There was a slight misunderstanding. With our calculations of the sensitivity factor $(R-R_H)^2$, we showed that higher-order Kiessig fringes are less sensitive than those closer to the critical edge for total reflection. The referee pointed out that “sensitivity to hydrogen mostly comes from the first and second

fringe", which completely agrees with our statement. We reiterate that we agree with the reviewer 1 that TOF reflectometers have a certain advantage in applying RNR for kinetical studies.

Reviewer #3 (Remarks to the Author):

Unfortunatly, I still disagree with the authors that RNR is required to detect hydrogen loading of very small amounts in thin films.

I have chosen to do NR calculation with the following two models.

- 1) Pt(3)/Nb(55)/Si and
- 2) Pt(3)/Nb(55)/Pt(10)/Si (their wave guide structure model)

Both with and without 5% H loading.

I assume the following parameters.

Sld Nb = 3.92, Si = 2.068, Pt (6.36), Sld Nb₉₅H₅ = 3.643 [Here I assume that density of Nb is 95% of pure Nb]

All of the interface roughnesses in both models are kept at 5Ang.

Here are the results that I get for T_{min} using their Eq. 6 and assuming beam intensity to be 1500/sec.

For the Pt(3)/Nb(55)/Si model I get T_{min}= .0392sec @ Q=.0155.

For the Pt(3)/Nb(55)/Pt(10)/Si model I get T_{min}= .0144sec @Q=.0155

I am not quite sure how the authors get a T_{min} of .25sec shown in Fig. 4c. I would urge them to check their calculation again. I have checked it several times and I am quite sure T_{min} of .0392sec is correct.

In any case the difference in the time between to cases is a factor of 3 and not 6 as claimed by the authors.

In the second case the total thickness is 10nm larger so the whole NR curve is shifted towards lower Q and that is one reason that T_{min} shifts to lower values.

I have checked this by increasing the thickness of the Nb film in the first model to 65nm (Pt(3)/Nb(65)/Si) and now T_{min}= .0223sec at Q=.0153. This is only 50% higher than model 2.

My sense after looking at this issue for some time is that whether one has a wave guide structure for RNR or the first model with only Nb film on Si, in both cases the measurement is being done at a low enough Q where the reflectivity is quite high. When reflectivity is already this high it is difficult to get gains of order of magnitude. The trick is to make a structure where the first minimum occurs at low enough Q and that can be easily done with NR.

If the authors disagree with my calculations, I will be happy to provide them with detailed sld profiles and calculated reflectivities.

We thank reviewer 3 for his critical review of our work. We checked our calculations once again and did not find any mistakes. The principal reason for the residual disagreement of both estimations of the minimum time is that the reviewer has used a different set of parameters for his new calculations. In addition, Reviewer 3 did not apply a correction for the momentum resolution, which is an additional (albeit smaller) source of discrepancy.

Our calculations Fig.4c (resolution $\Delta Q = 10^{-5} \text{ \AA}^{-1}$):

Name	d (Å)	σ (Å)	$\rho_{pristine}$ (10^{-6} \AA^{-2})	ρ_H (10^{-6} \AA^{-2})
Pt	30	5	5.7	
Nb	550	5	3.91	3.81
Si	∞	5	2.07	

Reviewer 3's calculations Fig.4c (no resolution correction):

Name	d (Å)	σ (Å)	$\rho_{pristine}$ (10^{-6} \AA^{-2})	ρ_H (10^{-6} \AA^{-2})
Pt	30	5	6.36	
Nb	550	5	3.91	3.64
Si	∞	5	2.07	

While we used a SLD of $3.81 \times 10^{-6} \text{ \AA}^{-2}$ for 5%-hydrogenated Nb, Reviewer 3 used the much smaller value of $3.643 \times 10^{-6} \text{ \AA}^{-2}$. The much larger difference to pristine Nb leads to a faster acquisition time. However, prior work has shown that a 5% decrease in density corresponds to 50 at.-% of absorbed hydrogen, or $\text{NbH}_{0.5}$ (see Ref. [30]), whereas our value is appropriate for $\text{NbH}_{0.05}$, the system we simulating. To avoid any ambiguity, we have quoted the SLD values we have used in the caption of Fig. 4.

Whatever value of the SLD one uses, RNR holds a significant advantage, as the reviewer himself has pointed out. The numerical difference between the calculations is therefore immaterial to our case for publication.